# Monitoring cropland daily carbon dioxide exchange at field scales with Sentinel-2 satellite imagery

Pia Gottschalk[1], Aram Kalhori[1], Zhan Li[1,2], Christian Wille[1], Torsten Sachs[1,3]

[1]GFZ German Research Centre for Geosciences, Potsdam, Germany

[2]Present address: BASF Digital Farming GmbH, Köln, Germany

[3] Institute of Geoecology, Technische Universität Braunschweig, 38106 Braunschweig, Germany

*Correspondence to*: Pia Gottschalk (pia.gottschalk@gfz-potsdam.de) and Torsten Sachs (torsten.sachs@gfz-potsdam.de)

**Abstract**

Improving the accuracy of monitoring cropland $CO_2$ exchange at heterogeneous spatial scales is of great importance to reducing spatial and temporal uncertainty in estimating terrestrial carbon (C) dynamics. In this study, an approach to estimate daily cropland C fluxes is developed and tested by combining time series of field scale eddy covariance (EC) $CO_2$ flux data

and Sentinel-2-satellite-based vegetation indices (VIs) after appropriately accounting for the spatial alignment between the two time series datasets. The study was carried out for an agricultural field (118 ha) in the lowlands of north-eastern Germany. The ability of different VIs to estimate daily net ecosystem exchange (NEE) and gross primary productivity (GPP) based on linear regression models was assessed. Most VIs showed high (>0.9) and statistically significant (p<0.001) correlations with GPP and NEE although some VIs deviated from the seasonal pattern of $CO_2$ exchange. In contrast, correlations between

ecosystem respiration (Reco) and VIs were weak and statistically not significant, and no attempt was made to estimate Reco from VIs. Linear regression models explained generally more than 80% and 70% of the variability of NEE and GPP, respectively, with high variability amongst the individual VIs. The performance in estimating daily C fluxes varied amongst VIs depending on C flux component (NEE or GPP) and observation period. Root mean square error (RMSE) values ranged from 1.35 g C m$^{-2}$ d$^{-1}$ using the Green Normilized Difference Vegetation Index (GNDVI) for NEE to 5 g C m$^{-2}$ d$^{-1}$ using the

Simple Ratio (SR) for GPP. This equated to an underestimated net C uptake of only 41 g C m$^{-2}$ (18%) and an overestimation of gross C uptake of 854 g C m$^{-2}$ (73%). Differences between measured and estimated C fluxes were mainly explained by the diversion of the C flux and VI signal during winter, when C uptake stayed low while VI values indicated an increased C uptake due to relatively high crop leaf area. Overall, results exhibited similar error margins as mechanistic crop models. Thus, they indicated suitability and expandability of the proposed approach to monitor cropland C exchange with satellite-derived VIs.

Key words: net ecosystem exchange, gross primary production, ecosystem respiration, satellite-derived vegetation indices, NDVI, EVI, EVI2, GNDVI, SAVI, SR, S2REP

## 1 Introduction

Managed cropland soils extend over 15.1-18.8 Mkm$^2$ (11.6 – 14.4% of the global ice-free land area) (Luyssaert et al., 2014). They store about 131.81 Pg of organic carbon (C) in the first 30 cm of the soil profile (Zomer et al., 2017) and constitute about 10% of the total terrestrial soil organic C stock (Jobbagy and Jackson, 2000). Cropland soils have historically lost a large amount of the original soil organic C (Guo and Gifford, 2002; Sanderman et al., 2017). This deficit, however, presents a large potential for sequestering C now and in the future (Lal et al., 2018; Zomer et al., 2017). Therefore, croplands have been

identified as the most promising land use type to compensate for fossil fuel emissions ('4 per mille' initiative (Minasny et al., 2017; Rumpel et al., 2020). Whether cropland soils are a net C source or sink though is determined by the total cropland C balance. As opposed to natural ecosystems in which the net C balance is mainly determined by the balance between gross primary production (GPP) and ecosystem respiration (Reco) (Chapin et al., 2006) only, the cropland net ecosystem C balance includes (lateral) C fluxes from harvest exports and manure imports (Ciais et al., 2010) and some minor C losses from C

leaching, erosion or fire. However, the atmospheric exchange of $CO_2$ of croplands (GPP & Reco) are the two largest and most uncertain fluxes in the regional cropland carbon balance (Ciais et al., 2010). Regionally integrated estimates of GPP and Reco are difficult in highly diverse and geographically patchy croplands, which results in high uncertainty of spatially-explicit estimates of cropland C stock changes (Pique et al., 2020). A robust knowledge and dedicated monitoring of the delicate balance between these two fluxes, however, is important to guide climate change mitigation measures. Furthermore, mitigation

measures based on cropland soil C sequestration require high accuracy in C flux estimates for monitoring, reporting and verification purposes.

The state-of-the-art method to measure ecosystem-atmosphere C exchanges is the *eddy-covariance* (EC) method based on micrometeorological theory (Baldocchi, 2003). This method allows for direct net ecosystem exchange (NEE) measurements which integrate C dynamics of spatially highly variable soil organic carbon (SOC) stocks.Subsequent flux processing partitions

NEE into GPP and Reco (Reichstein et al., 2005; Lasslop et al., 2010; Wutzler et al., 2018). Although respective results are robust and commonly accepted, they are confined to local, homogeneous footprint (FP) areas (Smith et al., 2010).

To assess and monitor the spatial variability of C fluxes at local scale and across ecosystems, a global network of EC flux sites (Fluxnet) has been established of which however, only 20 out of 212 sites (here: FLUXNET2015 Dataset) are cropland sites (Pastorello et al., 2020). They are thus sparse relative to the vast diversity of existing croplands. To overcome the spatial gap
between local measurements from a limited number of sites and regional to global C exchange estimates, the combination of local EC data with remote sensing products such as satellite-derived vegetation indices (VIs), has been explored (Tramontana et al., 2016; Jung et al., 2011; Fu et al., 2014; Bazzi et al., 2024; Mahadevan et al., 2008; Xiao et al., 2008; Xiao et al., 2010; Xiao et al., 2011).

The light-use-efficiency (LUE) concept (Medlyn, 1998; Yuan et al., 2014) and the relationship of fractional absorbed
photosynthetically active radiation (FPAR) with VIs (Myneni and Williams, 1994) allow VIs to be used as proxies for GPP (Running et al., 2004; Zhou et al., 2014). However, GPP is only one part of the C exchange and for estimating the full C budgets of ecosystem, the net exchange of C fluxes (NEE) is required. Therefore, the correlation of GPP with Reco (Baldocchi, 2008; Baldocchi et al., 2015; Ma et al., 2016) can be leveraged to directly link VIs with NEE (Noumonvi et al., 2019; Wohlfahrt et al., 2010; Huang et al., 2019b).

Interestingly, a direct correlation between GPP and VIs, rather than more complex approaches incorporating additional environmental drivers can outperform LUE type models such as the MODIS GPP ("MOD17") product across ecosystems (Sims et al., 2006), and particularly for croplands (Huang et al., 2019b). The high variability of green biomass during the phenological cycle seems to make croplands especially suitable for directly tracking GPP with VIs, without the need to incorporate meteorological drivers (Tramontana et al., 2015). The latter is attributed to the variability of plant dynamics being
determined rather by human interventions such as fertilizing, tilling, sowing and harvest dates (Tramontana et al., 2016). The goal of optimizing specific plant performance locally can, to some extent, override environmental conditions. This raises the question of whether a simple relationship between VIs and NEE can provide sufficient accuracy for estimating C fluxes in croplands, as opposed to more complex approaches.

While numerous studies assess the direct link of GPP with satellite-derived VIs (e.g. (Badgley et al., 2017; Huang et al., 2019b;
Peng and Gitelson, 2012; Joiner et al., 2018; Liu et al., 2021; Rahman et al., 2005; Wang et al., 2004; Juszczak et al., 2018; Lin et al., 2019) the number of studies assessing the correlation of VIs with NEE is small (Olofsson et al., 2008; Noumonvi et

al., 2019; Wohlfahrt et al., 2010; Huang et al., 2019b; Sims et al., 2006) and to our knowledge there is no such study including or dedicated to croplands.

With the increasing availability of higher resolution satellite imagery such as EnMAP, Sentinel-2C and Landsat Next, the link between these data and field scale C fluxes can provide greater spatial accuracy and requires further investigation. Except for the study by Madugundu et al. (2017), most studies exploring the potential of satellite-derived VIs to serve as proxies for estimating cropland C fluxes use MODIS or MODIS-like resolution products. Although the potential of finer-resolution satellite images such as Landsat or Sentinel-2, as opposed to coarser, MODIS-like resolution, for cropland C flux estimates at spatial scales has been demonstrated (Gitelson et al., 2012; Fu et al., 2014; Wolanin et al., 2019; Bazzi et al., 2024; Madugundu et al., 2017; Chen et al., 2010; Spinosa et al., 2023; Pabon-Moreno et al., 2022) a direct link between VIs and cropland NEE has not been explored. All these studies employ complex approaches (process-based models, machine learning) which demand high computational efforts and are heavy on auxiliary data requirements. Further, complex approaches always introduce additional sources of uncertainty: model structure and parameter uncertainty and input data uncertainty (Wattenbach et al., 2006) and can be difficult to parameterise (Sims et al., 2006). Exploring a direct and straightforward link would clarify the explanatory power of VIs themselves, enabling systematic examination of the conditions under which additional data are truly necessary without undermining the comprehensive meaningfulness of the VIs. Furthermore, except for Bazzi et al. (2024) and Fu et al. (2014), all studies focus on GPP rather than NEE and most of the mentioned studies explore only a few VIs. Additionally, none of these studies are dedicated to examining the link between C fluxes and VIs along the phenological cycle at plot scale which is imperative to evaluate the robustness and accuracy of linking the two signals.

Another advantage of using finer resolution imagery is the ability to match the spatial footprint of the two signals. Large area products of C fluxes which combine remotely-sensed VIs with EC observed C dynamics (Jung et al., 2019) (the FLUXCOM initiative: http://www.fluxcom.org/) can have a relatively low spatial resolution of 0.0833°. Such a coarse resolution can cause a systematic mismatch between the satellite sensor and EC tower FP (Tramontana et al., 2016) and mostly does not well distinguish individual agricultural fields. To improve the estimation of C exchange for croplands, the need for higher resolution remote sensing data such as Landsat has been pointed out previously (Tramontana et al., 2016). Furthermore, Kong et al.

(2022) showed the superiority of FP matched regression between GPP and high-resolution satellite near-infrared (NIR) maps over *in situ* (NIR sensor location) regressions for different cropping fields in California.

All of this highlights the need and shows promise to further investigate the capabilities of high-resolution satellite imageries in directly estimating carbon fluxes in croplands and monitoring them by a rigorous evaluation of comprehensive spectral 110 indices derived from finer-resolution satellite imageries, while appropriately leveraging EC-based C flux measurements.

The most commonly used VI in combination with NEE is the normalized difference vegetation index (NDVI), followed by the enhanced vegetation index (EVI) and the land surface water index (LSWI). Noumonvi et al. (2019) additionally used the green NDVI (GNDVI), the normalized difference surface water index (NDSWI), the soil-adjusted VI (SAVI), and the modified 115 normalized difference water index (MNDWI). Wohlfahrt et al. (2010) use the simple ratio (SR) and Tramontana et al. (2016) the normalized difference water index (NDWI). However, only Tramontana et al. (2016) link VIs (NDVI, EVI, NDWI and LSWI) to cropland NEE, GPP and Reco while the other studies assess their suitability for grassland C fluxes. Assuming the predictive performance of GPP from VIs for croplands also holds true for cropland NEE and Reco, our VI selection was based on VI performance for GPP estimation at croplands. In this regard typical satellite-sensor derivable VIs such as NDVI, EVI, 120 EVI2, and SR (Huang et al., 2019a; Peng and Gitelson, 2012) indicate the highest potential for our study.

Here, we introduce a new agricultural EC measurement site in north-eastern Germany and present daily and annual $CO_2$ dynamics over two and a half growing seasons along with the relevant site, meteorological and management data. We explore the capacity of high-resolution satellite imagery in conjunction with a comprehensive range of VIs to estimate daily NEE, GPP 125 and Reco. To overcome the problem of the potential spatial mismatch of the two signals, the source area of the two signals was matched to the same footprint area. Instead of average NEE values (e.g. midday or 8-day averages around acquisition dates), integrated daily NEE values were used to allow for continuous full C budget calculations. Challenges along the course of the phenological cycle were analyzed to better understand the achievable accuracy and uncertainties associated with this approach.

To summarise, the objectives of this paper thus were (1) to present and evaluate the $CO_2$ dynamics and C budgets of a newly established EC cropland site in north-eastern Germany, (2) to assess the performance of a range of various, high resolution imagery derived VIs to estimate daily NEE, GPP and Reco at this respective site, and (3) to discuss and evaluate the results of this simple approach in comparison to more complex methods and future research requirements.

## 2 Data and methods

### 2.1 Site description

The net ecosystem exchange is measured with an eddy covariance (EC) system located on an arable field in the north-eastern lowlands of Germany (53°52'05.7"N 13°16'07.0"E), southeast of the village of Heydenhof (Figure 1). The site is situated in an upper pleistocene landscape with a temperate-oceanic climate of yearly mean temperature and precipitation of 8 °C and 580 mm for 1961-1990 and 9.2 °C and 575 mm for 1991-2020, respectively (nearby German Weather Service climate station "Anklam" and "Greifswald"). The soil is a clayey loam with more than 10% clay and contains about 1.5-2% organic C with a high gradient across the field (pers. comm. with land manager, 19 Jan 2022). The continuous half-hourly and ongoing EC measurements started on 4 Mar 2020, 17:30 UTC at a measurement height of seven meters above ground. The site has been under arable cropping for at least 60 years with a crop rotation of 1/3 winter rape (WR) and 2/3 winter wheat (WW) during the last years (pers. comm. with land manager, 19 Jan 2022). The details of the crop rotation and management for the time and field under investigation surrounding the EC tower (named "main field" here) is outlined in Table 1. Exact dates of management were not provided by the land manager and were delineated from visual inspection of half-hourly imagery provided by a tower-mounted camera overlooking the field facing northwards.

The EC tower is situated close to the border of an adjacent field to the east (distance: 120 m) and relatively close to an adjacent field to the south (distance: 285 m) (Figure 1). The tower itself sits on the southern edge of a dry cattle hole (north-south length 45 m, width 25) at an altitude of 22.7 m above sea level.

**Table 1** Crop management information for the "main field". Note, information about yield and straw amounts and crop and soil management is based on long-term averages. Yearly field scale data are confidential to the farmer.

| Season | 2019/2020 | 2020/2021 | 2021/2022    155 |
|---|---|---|---|
| Crop | winter rape | winter wheat | winter wheat |
| Sowing date | approx. 12 Aug to 25 Aug | approx. 15 Sep to 15 Oct | 14 Oct 2021 |
| Harvest date | 23 Jul 2020 | 09 Aug 2021 | 21 Jul 2022 |
| Yield [t C ha$^{-1}$] | 1.8 | 4.05 | 4.05 |
| Straw [t C ha$^{-1}$] | | 1.92 (left in the field) | 1.92 (removed) |
| Fertilizer [kg N ha$^{-1}$] | 100 (cattle manure in autumn) + 140 (urea in spring) | 220 (urea in spring) | 220 (urea in spring), 160 partly replaced by organic fert. |
| soil cultivation/ploughing | 3x between harvest and sowing | 3x between harvest and sowing | 3x between harvest and sowing |
| Herbicides/pesticides | 3-4x in spring | 4x in spring | 4x in spring |

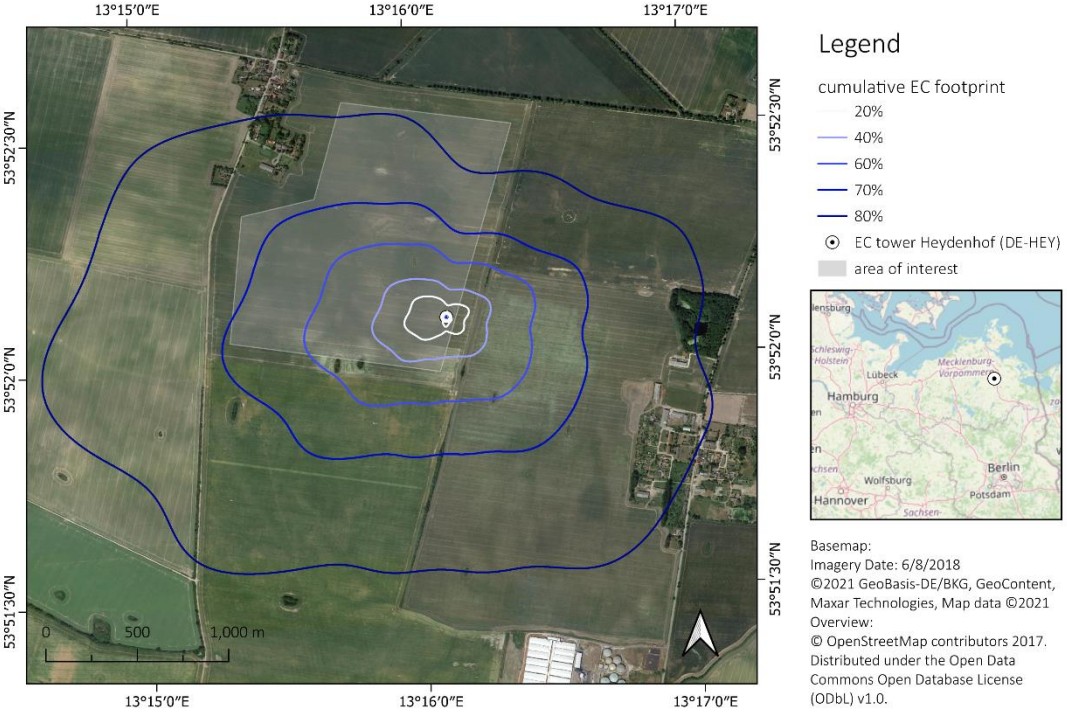

**Figure 1** Setting and layout of the arable fields surrounding the EC tower in north-eastern Germany. Isolines denote the cumulative contribution of the source area to the flux signal over the measurement period (05 Mar 2020 to 23 Aug 2022) of the "Heydenhof" EC tower. For more detailed information of the cumulative source area please refer to Appendix B. The transparent grey area outlines the polygon for which average satellite-derived vegetation indices were calculated. The outer borders of the respective field (as seen on the picture) surround the area for which the homogeneous (in terms of crop dynamics) EC flux time series is calculated ("main field", see text). [Original map designed by Karl Kemper.]

## 2.2 Measurement equipment and raw flux data processing

EC flux measurements were carried out with a 3D ultrasonic anemometer (HS-50, Gill Instruments, UK) and an open-path infrared gas analyzer (LI-7500DS, LI-COR Biosciences, USA). Data from these sensors were measured with a frequency of 20 Hz. Half-hourly fluxes were calculated with the software EddyPro (version 7.0.7, LI-COR Biosciences, USA). Meteorological data including air temperature and relative humidity (HMP155, Vaisala, FI), barometric pressure (Model 61302V, Young, USA), and incoming and outgoing shortwave and infrared radiation (CNR4, Kipp & Zonen, NL) and photon

flux density (LI-190, LI-COR Biosciences, USA) were measured with a frequency of 1 Hz and averaged to half-hourly values. 15-min-precipitation data were collected with an RG Pro Adcon Rain Gauge (Itzerott et al., 2018).

## 2.3 EC data processing

The handling of half-hourly NEE data, from 5 Mar 2020 to 23 Aug 2022, followed standard EC data processing steps and are

further detailed in appendices A-C. Data were quality controlled (Appendix A) and filtered for spatial representation of the "main field" by foot-print modelling (Appendix B). The FP filter-threshold was optimized according to data availability (i.e. acceptable number of gaps) and representativeness of the 'main field'. Times of insufficient turbulence were filtered by the u*-threshold approach (Appendix C) and subsequent gap-filling (Appendix C) used only data of highest quality, i.e. quality flag = 0 following the CarboEurope IP flag convention (Mauder and Foken, 2004). Flux partitioning into GPP and Reco

followed Reichstein et al. (2005).

Half-hourly C fluxes were subsequently aggregated to daily sums for linking with satellite data. General data processing, analysis and visualization was carried out in R (R Core Team, 2021). NEE sign notation followed the micrometeorological sign convention, which denotes C gains by the vegetation with negative values and C losses to the atmosphere from auto- and heterotrophic respiration with positive values (Aubinet et al., 2009). When describing and discussing C fluxes in the text, NEE

and GPP are referred to by their absolute values, such that NEE and GPP "decrease" with a decrease in C uptake.

**2.4 C budget calculation and evaluation**

To assess the magnitude of the C exchange as compared to the other components of the cropland soil C budget, a simplified C budget (A4) for the two WW growing seasons (sowing to harvest, Table 1) was calculated as follows:

C budget = NEE – import + export

where import is limited to seeds' C since no manure is applied and export refers to C harvest losses.

**2.5 Satellite-based vegetation indices**

Average values of VIs (Table 2) of satellite imagery pixels were calculated from within the borders of the 'main field' (transparent polygon in Figure 1). The source area of the satellite signal thus matches with the source area of the EC tower.

The L2A products of Sentinel-2 multi-spectral instrument, provided by the Copernicus program of the EU and the European Space Agency was used. Sentinel-2 image processing was carried out at the Google Earth Engine platform (Gorelick et al., 2017). The quality map SCL (scene classification) of the Sentinel-2 L2A product is used to filter cloud, cloud-shadow and saturated pixels to ensure only high-quality scenes were selected for the calculation of the vegetation indices. Satellite overpass time is approximately 10 am (local solar time) for Heydenhof.

A continuous time series of daily satellite data was constructed by linear interpolation for the days between acquisition dates.

**Table 2** List of vegetation indices calculated from Sentinel-2. RED, GREEN, BLUE, NIR (near-infrared) and SWIR (shortwave infrared) refer to the respective spectral bands of Sentinel-2.

| Index | Formula | Name, range, purpose |
|---|---|---|
| NDVI | (NIR - RED) / (NIR + RED) | *normalized difference vegetation index*: characterizes the density/green biomass of vegetation (Rouse et al., 1974) |
| EVI | 2.5*((NIR - RED) / (NIR + 6 * RED - 7.5 * BLUE + 1)) | *enhanced vegetation index*: reducing soil and atmospheric contamination of vegetation signals and optimizing the vegetation signal in areas with a high leaf area index (LAI) where NDVI would saturate (Liu and Huete, 1995) |
| EVI2 | 2.5*((NIR - RED) / (NIR + 2.4 * RED + 1)) | *2-band enhanced vegetation index*: an index that has the best similarity with the EVI but without using blue band at which surface reflectance values can be sensitive to residual errors in atmospheric correction (Jiang et al., 2008) |
| GNDVI (= - NDWI$_{McFeeters}$) | (NIR - GREEN) / (NIR + GREEN) | *green NDVI*: indicator of the photosynthetic activity of vegetation assessing the moisture content and nitrogen concentration in plant leaves, more sensitive to chlorophyll concentration than NDVI (Gitelson et al., 1996); the GNDVI is the inverse of the NDWI as defined by Mcfeeters (1996) |
| NDSVI | (SWIR1 - RED) / (SWIR1 + RED) | *Senescence index (0-1)*: to detect senescent vegetation (Qi et al., 2002) |
| NDWI = LSWI | (NIR - SWIR1) / (NIR + SWIR1) | *normalized difference water index/ land surface water index (-1 – 1)*: used to monitor changes in water content of leaves, should be used complementary to NDVI, not to substitute NDVI (Chandrasekar et al., 2010; Gao, 1996) |
| MNDWI | (GREEN-SWIR1)/(GREEN+SWIR1) | *modified normalized difference water index:* modified from NDWI$_{McFeeters}$ to enhance open water features (Xu, 2006) |
| SAVI | [(NIR-RED)/(NIR+RED+L)]*(1+L), L=0.5 | *soil-adjusted vegetation index*: this transformation of the NDVI nearly eliminates soil-induced variations in the VI (Huete, 1988) |
| SR | NIR/RED | *simple ratio*: leaves absorb relatively more red than infrared light, thus, the ratio increases with more green biomass (Jordan, 1969) |
| S2REP | 705+35*((((RE3+RED)/2)-RE1)/ (RE2-RE1)) | *Sentinel-2 red-edge position (REP)*: REP is the point of maximum slope along the RE; has been used to enhance |

| | | estimates of leaf and canopy chlorophyll content (Frampton et al., 2013; Guyot and Baret, 1988) |
| --- | --- | --- |

**2.6 Correlation between daily C fluxes and vegetation indices**

Correlations and linear regressions between daily C fluxes and VIs were calculated for the days at which reliable satellite data were available. The correlation was used to identify which VIs were most suitable for estimating C fluxes. Higher correlations indicate a higher coincidence and thus higher suitability of a VI to estimate C fluxes.

**2.7 Estimation and evaluation of daily C fluxes from VIs by linear regression**

Simple linear regressions of the type

$$C\ flux = a*VI+b \hspace{4cm} \text{Equation 1}$$

were fitted to the 73 data pairs to subsequently estimate daily C flux values from interpolated satellite data. Resulting linear regressions were tested for statistical significance and the coefficient of determination ($R^2$) indicates the amount of variability in the dependent variable (i.e. C flux) explained by the regression.

As a measure of accuracy for the final estimation of daily C fluxes (902 data points), the correlation coefficient ρ was used for association (trend similarity) and the $R^2$ and RMSE for coincidence, i.e. percentage variability explained by the regression and total difference between measured and estimated value, respectively. Additionally, the "model efficiency" factor E (Nash and Sutcliffe, 1970) was calculated to characterize the ability of the linear models to replicate daily C fluxes. Values range from -∞ to 1 where 1 indicates a near perfect fit, 0 denotes that the approach is not better than taking the mean of the observations and any value below 0 rates the estimation approach as poor.

Linear models were set up for three different evaluation periods: 1) the whole observation period ranging from first to last satellite image, 2) for the two WW growing periods ranging each from sowing to harvest, and 3) for the first WW growing period (WW1). The latter was subsequently used to estimate daily C fluxes of the second WW growing period (WW2) as an evaluation of the temporal transferability of this approach and associated absolute errors. Our linear regression-based WW2 C flux estimates were subsequently evaluated by comparing our results to simulation results of mechanistic crop models and to satellite data-model fusion approaches estimating cropland C fluxes at various (other) sites.

## 3 Results and discussion

### 3.1 Evaluation of C exchange dynamics and C budgets

In total, 902 days of half-hourly flux data contributed to this analysis. Since only measurements of highest quality were kept for subsequent processing (see section 2.3), only 44% of half-hourly data were available for further processing.

This was further reduced by the integration of the FP model results. Filtering the dataset for the "main field" (FP modelling) reduced the available "qc0-data" to 18.7% and 12.8% for a FP filter threshold of 0.7 and 0.8, respectively. The higher threshold renders very long and continuous gaps during winter, which makes the gap-filling highly uncertain and was the main reason for the deviations between the two gap-filled time series using 0.7 or 0.8 as the threshold for FP filtering (data not shown). While using the FP model threshold of 0.8 valid fluxes might have been more representative for the "main field", the time series employing the threshold of 0.7 was concluded to be more reliable and well balancing the loss of some representativeness. Still, available data coverage was quite low compared to other studies (Schmidt et al., 2012). The proportion of missing data is higher during nighttime (defined as fluxes at $<20$ W m$^{-2}$ global radiation) than during daytime, 93% versus 70% respectively (for the threshold of 0.7).

The final time series was gap-filled using a single u*-threshold of 0.2 as estimated by REddyProc (Wutzler et al., 2018). The large proportion of gaps did not allow for season specific u*-threshold estimation. Moureaux et al. (2008) also used a single value for a WW crop in Belgium which was very similar to ours, 0.22. They further suggested that u*-uncertainty had a very low impact on fluxes. Figure 2 depicts the gap-filled NEE, GPP and Reco curves along with relevant meteorological variables.

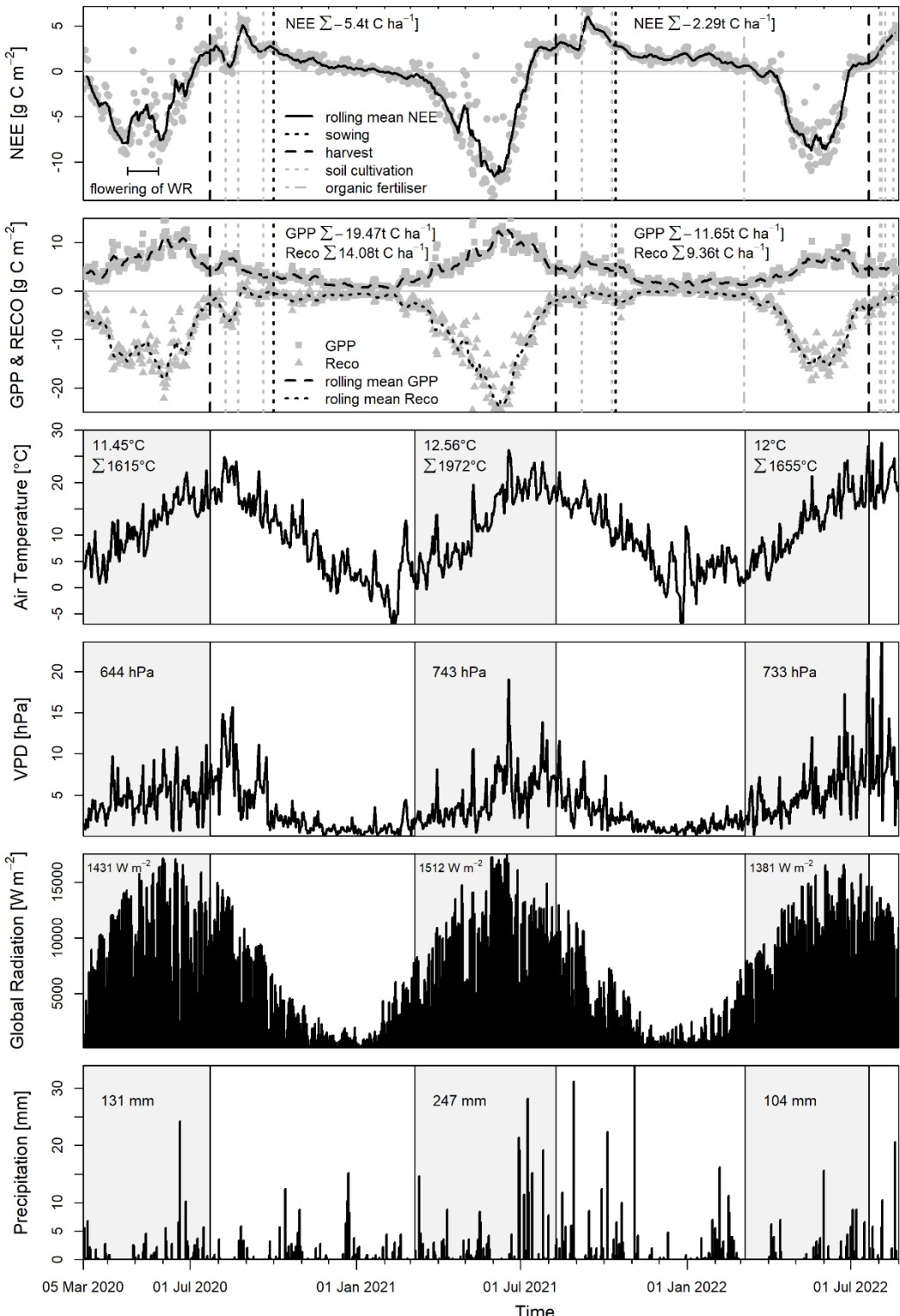

**Figure 2** Gap-filled timeseries of daily sums of NEE (grey dots), GPP (grey squares) and Reco (grey triangles) [g C m$^{-2}$ d$^{-1}$], 10-day rolling means of NEE(solid black line in top panel), GPP (dashed black line) and Reco (dotted black line) and auxiliary meteorological variables (daily mean air temperature (Tair) [°C], daily mean vapor pressure deficit (VPD) [hPa], daily sum of global radiation (Rg) [W m$^{-2}$], daily sum of precipitation [mm]). Numbers in the C flux panels denote the seasonal (sowing to harvest) cumulative C fluxes. "Flowering of WR" in the top panel illustrates the duration of the flowering of winter rape (WR) from 23 Apr to 27 May. Grey boxes in the meteorological data panels denote the time period of the "climate growing season" for which descriptive climate parameters were calculated. They start with the 05 Mar each year to match the first growing season in which EC measurements only started on 05 Mar 2020. They end on the days of harvest each year: 23 Jul 2020, 9 Aug 2021 and 21 Jul 2022. Dates for field management actions were determined from the inspection of tower mounted field camera photos since the respective information was not given by the farmer. Numbers in the grey boxes denote mean air temperature, temperature sums with base temperature 0, sum of VPD, Rg and precipitation for the climate growing seasons, respectively.

Generally, growing conditions were more favorable in the second growing season than in the first and last (see "climate growing season" parameters in Figure 2) which was reflected in the higher absolute values of all fluxes in 2021 than in 2020 and 2022. Overall, NEE, GPP and Reco flux dynamics at our cropland site showed a typical pattern of European WR and WW cropping. An in-depth description of C flux dynamics at our cropland site is presented in S1.

The total C budget during the observation period accumulated to a total loss of 4.46 t C ha$^{-1}$ accounting for C exports through WR- and WW-harvests and straw removal after the last harvest, ignoring C import via seeds. However, C import via seeds typically ranges between 0.02-0.08 t C ha$^{-1}$ for WW (Aubinet et al., 2009; Schmidt et al., 2012; Waldo et al., 2016) which is negligible compared to the other C budget components. C export through harvest and straw removal amounts to 11.82 t C ha$^{-1}$. C losses from the systems thus outbalance the total net uptake of -7.36 t C ha$^{-1}$. Uncertainty as provided by the u*-bootstrapping procedure of REddyProc ranged from -7.53 t C ha$^{-1}$ (0.05 percentile) to -7.37 t C ha$^{-1}$ (0.95 percentile) for the net atmospheric C exchange. The C budget of the two individual WW seasons amounted to a net C uptake in 2020/2021 of -1.34 t C ha$^{-1}$ and to a net C loss in 2021/2022 of 3.68 t C ha$^{-1}$ due to the removal of straw. They were well within the range of

respective C budgets of European WW sites, ranging from -4.45 to 2.54 t C ha$^{-1}$ (Waldo et al., 2016; Anthoni et al., 2004; Aubinet et al., 2009; Béziat et al., 2009; Li et al., 2006; Schmidt et al., 2012; Wang et al., 2015). Individual C budget

components are reported in Table 3.

**Table 3** C budget components for the two WW growing seasons. Sums of C fluxes from sowing to harvest are reported in t C ha$^{-1}$. Values in brackets denote the 0.05 and 0.95 percentile from the uncertainty estimation based on bootstrapping (Wutzler et al., 2018).

| WW | NEE | GPP | Reco | harvest | straw | C-budget |
|---|---|---|---|---|---|---|
| 2020/2021 | -5.4 (-5.47 – -5.34) | -19.47 (-19.44 – -18.63) | 14.08 (13.17 – 14.1) | 4.05 | - | -1.34 (-1.42 – -1.29) |
| 2021/2022 | -2.29 (-2.292 – -2.3) | -11.65 (-11.68 – -11.35) | 9.36 (9.05 – 9.38) | 4.05 | 1.92 | 3.68 (3.67 – 3.68) |

**3.2 Satellite vegetation indices**

In total 73 scenes with good quality surface reflectance data were obtained for our 'main field' for the period 5 Mar 2020 to 23 Aug 2022. Standard deviations of VIs within the main field per each image were small indicating low spectral heterogeneity of the main field and were thus considered as negligible for the purpose of this study (Figure 3).

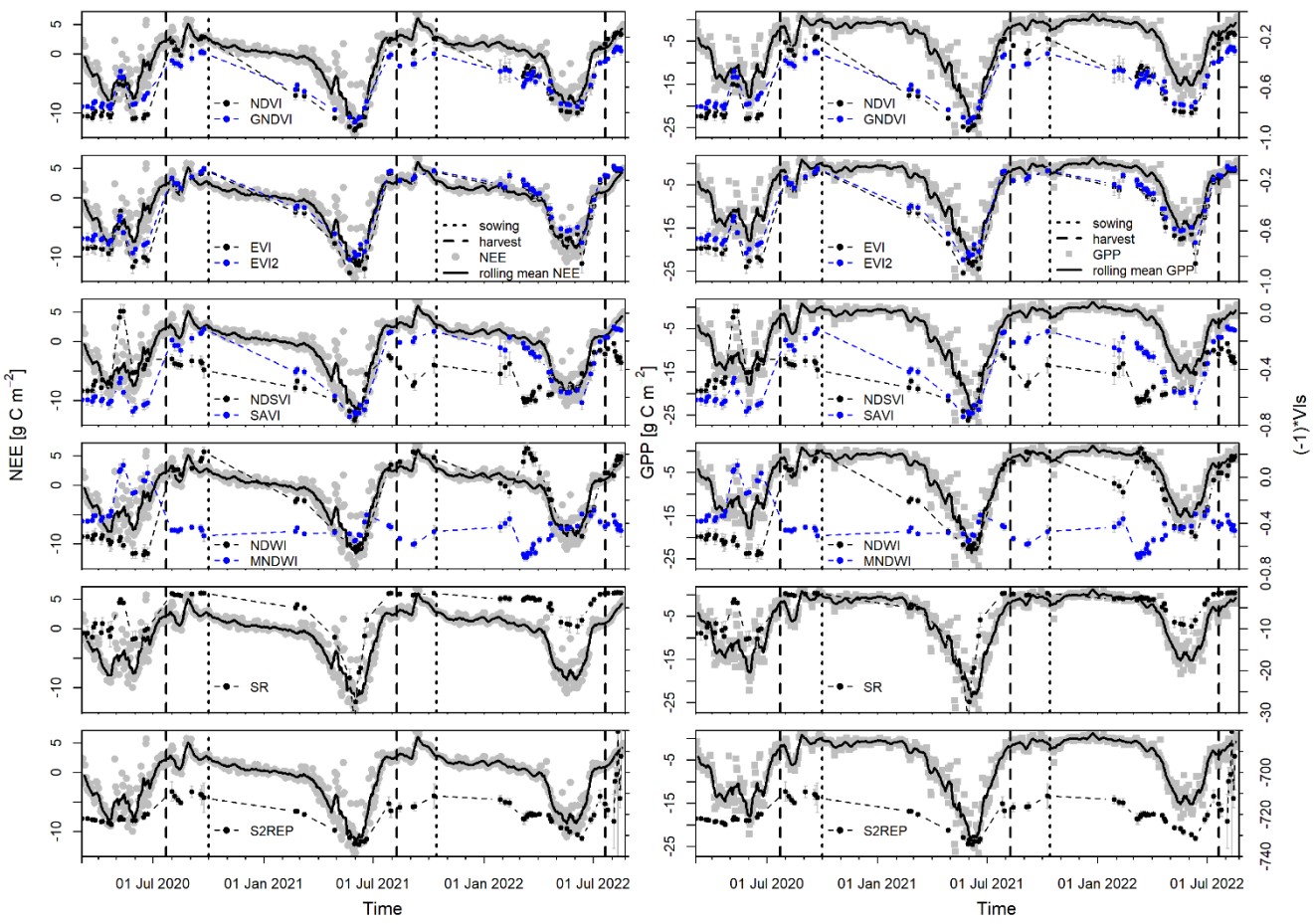

**Figure 3** Course of NEE (grey dots) and GPP (grey squares) plotted against Sentinel-2 derived vegetation indices. C flux measurements are aggregated to daily sums superimposed with a 10-day rolling mean (black curves). Note: vegetation indices (black and blue dots) are plotted inversely to facilitate the comparison of the dynamic pattern of the two types of signals. Error bars (grey) of VIs indicate the standard deviation across the 'main field'. Dashed black and blue lines connect 295 the individual VIs to facilitate visually the course of VIs over time.

The temporal dynamics of satellite-derived VIs generally matched well with the seasonal pattern of NEE and GPP (Figure 3) – except for from NDSVI and MNDWI – where an increase in absolute VI values concurred with an increase of absolute NEE and GPP values. One marked deviation occurred during the onset of the winter seasons which coincided with longer gaps in

satellite images (winter months are generally characterized by higher cloud cover causing gaps in satellite imagery time series). Here, the gap in winter VI data was characterized by a presumably initial increase of absolute VI values without indication of a maximum and by relatively high absolute VI values at the onset of spring in the following year. However, a respective dynamic (increase in absolute GPP or NEE) could not be observed. GPP values showed low C uptake (about -1 g C m$^{-2}$ d$^{-1}$) from October to February while net C uptake only slightly increased (due to decreasing Reco). The increase of absolute VI values towards the end of the years 2020 and 2021 for WW, which is not common (Itzerott and Kaden, 2006a, b), was observed at our site in the years prior to our study period as well (data not shown). Similar patterns had also been observed for WW in Kansas, US (Masialeti et al., 2010) and an immediate increase in "greenness" after crop emergence was also corroborated by the "Greenness Index" of the PhenoCam pictures of Heydenhof (https://phenocam.nau.edu/webcam/roi/heydenhof/AG_1000/). It was thus hypothesized that the diversion between the NEE and GPP signals and the VI signals can be explained in analogy to the phenological development of winter crops. After emergence the plants quickly developed a relatively high leaf-area-index (LAI) with a high specific leaf area (cm$^2$ g$^{-1}$) before winter dormancy. They assimilated less C into biomass (i.e. GPP) per leaf area than during the warmer part of the growing season (Korres et al., 2014; Van Oosterom and Acevedo, 1993; Weaver et al., 1994). Under light-limiting conditions, plants invest in leaf area rather than leaf biomass (Rawson et al., 1987). This presumably caused the mismatch between the course of GPP and NEE, and VIs. This winter deviation counteracted the straight forward linear correlation which was observed for the rest of the observation period. It denoted a systematic discrepancy of the linear relationship between VIs versus GPP and NEE. Note however, that Sentinel-2 VIs were able to pick up the decline of NEE and GPP during WR flowering (from around 23 Apr 2020 to around 27 May 2020). The sensitivity of VIs to WR flowering has also been reported by Itzerott and Kaden (2006b) for NDVI.

A less obvious deviation seemed to occur during senescence of WR. While the VI signals of NDVI, GNDVI, EVI, EVI2, SAVI and NDSVI pick up the senescence related drop in NEE for WW immediately, these VIs lag about 18 days (average across VIs based on the observation that VI values had reached a plateau at the same time as of max. C uptake (= min NEE) and did not drop at the same time as NEE decreases; another similarly high VI value is observed about 18 days later before the next VI value is much lower) behind the NEE signal for the senescence period of WR.

Most VIs were sensitive to the distinctly different C uptake dynamics of the summers of 2021 and 2022. C uptake is lower in 2022 and most VIs, except S2REP, MNDWI and NDSVI, reach respectively different levels of maximum absolute values.

### 3.3 Correlations between daily C fluxes and vegetation indices

Since the 73 data points of NEE, GPP and Reco were not normally distributed (Shapiro-Wilk normality test, $p < 0.05$), the non-parametric spearman's rank correlation coefficient $\rho$ was used to determine linear correlations. The linear correlations
(spearman's $\rho$) and respective significance levels among C fluxes and VIs were generally high and significant for NEE and GPP but low and less significant for Reco (Figure 4). Correlations among NEE, GPP and Reco were all significant ($p < 0.001$). GPP and Reco, and GPP and NEE had $\rho$-values of -0.8 and 0.88 respectively. Baldocchi (2008) and Baldocchi et al. (2015) reported correlations between GPP and Reco across ecosystems of 0.89 and -0.83, respectively. Please note, that Baldocchi (2008) reports GPP ('$F_A$') with a positive sign, not with a negative sign as we do and as in Baldocchi et al. (2015). Thus, his
correlation value between GPP and Reco is positive as opposed to ours and in Baldocchi et al. (2015). The correlations can still be compared based on the absolute values. NEE and Reco correlated by -0.47 only.

NEE and GPP correlated highly significant ($p < 0.001$) with all VIs except MNDWI which showed a lower level of significance ($p < 0.01$). GPP correlated highly significant ($p < 0.001$) with all VIs except MNDWI ($p < 0.01$) and NDSVI ($p < 0.05$) and both correlations showed a lower level of statistical significance. NEE correlated best with EVI2 and SAVI ($\rho = -0.93$), followed by
EVI, NDVI, GNDVI and SR ($\rho = -0.92$) and S2REP ($\rho = -0.9$), while NDWI showed a $\rho$-value of -0.82. NDSVI and MNDWI had lower correlations of -0.51 and -0.23 only. GPP generally showed lower correlations with VIs and a different ranking in the following order: EVI, EVI2 and SAVI with a $\rho$-value of -0.84, followed by NDWI (-0.83), S2REP (-0.81), NDVI (-0.8), SR (-0.79), GNDVI (-0.77) and MNDWI and NDSVI with -0.38 and -0.26 respectively. Reco showed significant but lower correlations with VIs except NDSVI with no significant correlation. The highest correlation was observed for NDWI with 0.58
followed by S2REP with 0.5 and EVI, EVI2 and SAVI with 0.47. The low correlation between Reco and VIs was not surprising though because respiration and its underlying auto- and heterotrophic processes do not directly connect with any surface reflectance characteristics (Wohlfahrt et al., 2010). While autotrophic respiration is closely linked with crop productivity (GPP) (Suleau et al., 2011) and has been shown to dominate total Reco in cropland systems during the growing season (60-90%) (Suleau et al., 2011; Zhang et al., 2013), the processes and environmental factors governing heterotrophic processes are more

complex (Grace et al., 2007), and not yet fully understood. While temperature had low explanatory power for autotrophic respiration, it is a strong driver of heterotrophic respiration (Suleau et al., 2011) which can dominate total Reco at an annual basis (Zhang et al., 2013). The other main factor influencing heterotrophic respiration rates is the availability of organic substrate, while soil moisture might have a negligible impact in this context when soil water content is between wilting point and field capacity (Zhang et al., 2013). We speculate that the low correlation between Reco and VIs is mainly due to the

heterotrophic component of Reco. Additional information on temperature and soil organic matter content could contribute to improving this simple approach, although data availability for separating heterotrophic and autotrophic respiration might be a limiting factor.

A correlation analysis using data points from the WW growing periods only, indicated very similar correlations which varied only by 0.1 or 0.2 but the overall pattern stayed the same (data not shown). Additionally, running the correlation analysis using

only half-hourly flux data for the time of satellite overpass, i.e. flux data of 10:00 - 10:30AM, the correlations of C fluxes with satellite VIs only slightly decreased in correlation intensity but again the overall pattern and significance levels stayed the same (data not shown). This supported the applicability of daily accumulated fluxes for the presented approach as the aim was to estimate total C exchange over time.

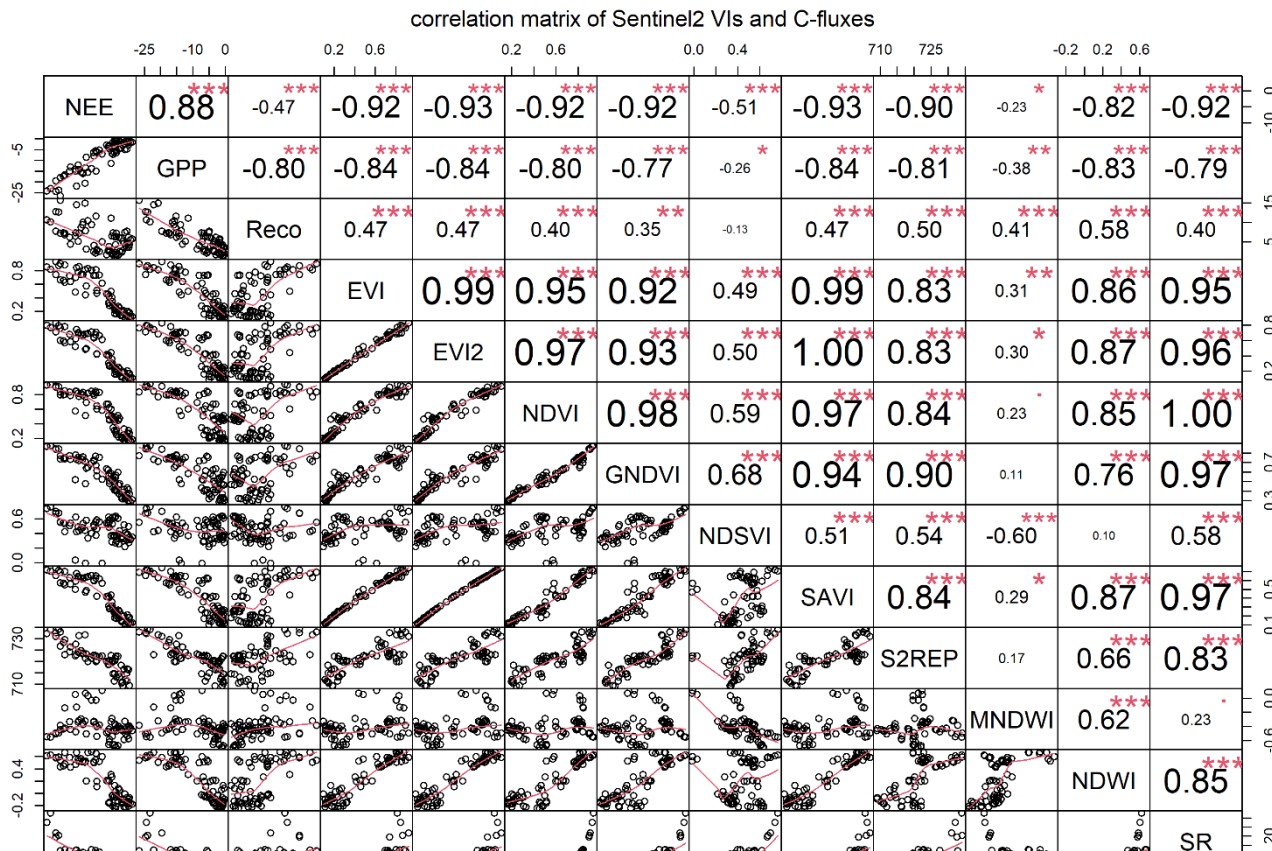


**Figure 4** Rank-correlation matrix of NEE, GPP and Reco with EVI, EVI2, NDVI, GNDVI, NDSVI, SAVI, S2REP, MNDWI, NDWI and SR from Sentinel-2. Numbers indicate the correlation coefficient ρ, the size of the number is scaled by the degree of correlation. Red stars denote the statistical significance levels, such as '***' p<0.001, '**' p<0.01, '*' p<0.05, '.' <0.1, ' ' <1. Scatter plots display a fitted line line (red).


Similarly high and significant correlations had been found between VIs and daily GPP/NEE for grasslands (Noumonvi et al., 2019; Wohlfahrt et al., 2010) and between VIs and daily GPP for WW crops (Juszczak et al., 2018). Noumonvi et al. (2019) also showed a lower correlation of NDSVI with NEE and GPP as compared to other VIs. While the correlations between VIs and daily GPP were found generally higher than with NEE for grasslands in previous studies (Noumonvi et al., 2019; Wohlfahrt

et al., 2010) as opposed to our results for croplands, Noumonvi et al. (2019) showed that the correlations with NEE during dry phases were higher than with GPP for nearly all the tested VIs. Their definition of "dry phase" (VPD > 1500 Pa) however did not apply to the weather conditions of our site for the time of observation. Further investigations are needed to understand the causes of differences in the efficacy of VIs to estimate GPP versus NEE and the impact of dry/wet conditions on the efficacy. Generally, our correlations for NEE and GPP were high compared to correlation of VIs with other crop vegetation parameters such as biomass, LAI, chlorophyll a and b or total nitrogen content (Boegh et al., 2002; Lilienthal, 2014) which are common parameters inferred from VIs for crop growth monitoring.

Overall, VIs based on the red, green, NIR or red edge spectral bands of the satellite sensors, such as NDVI, GNDVI, EVI, EVI2, SAVI, and SR showed better correlations with NEE than VIs containing short wave infrared (SWIR) band information, such as NDSVI, NDWI, and MNDWI. This did not hold true for correlations between VIs and GPP, where NDWI showed the second highest correlation with GPP. Reco even correlated relatively best with NDWI. VIs using red, green, NIR or red edge spectral bands were developed and are extensively used to evaluate the "greenness" or photosynthetic activity of plants. It was thus surprising that VIs correlated better with the NEE signal than with GPP which is more directly related to photosynthetic plant activity, while NEE is composed of the two opposing fluxes of GPP and Reco.

### 3.4 Linear models to estimate daily C fluxes

For the linear modelling the analyses were confined to NEE and GPP since correlations between VIs and Reco were comparably low and Reco could also be calculated by deducting GPP from NEE. The curved relationship of NEE and GPP, respectively, and VIs required a data transformation. EVI2 was chosen to determine the type of transformation because EVI2 showed one of the highest correlations with NEE (Figure 4). Here, transforming NEE values to log(-NEE+10) gave the best linear regression between EVI2 and NEE ($R^2$=0.86, p<0.001, residual standard error=0.14). GPP values were transformed likewise with log(GPP+10) ($R^2$=0.78, p<0.001, residual standard error=0.18).

Interception and slope parameters and the coefficient of determination ($R^2$-value) of VIs and NEE data were all statistically significant at the level of p<0.001 (Table 4).

**Table 4** Statistics of linear regression models of NEE and GPP versus VIs for 73 (whole observation period), 37 (both WW growing periods), 13 (first WW growing period – WW1) and 24 (second WW growing period – WW2) data pairs respectively. All parameters were statistically significant at the level of p<0.001, except where explicitly stated: p<0.01 ('**'). *Note: NEE and GPP were both log-transformed, see text. To increase readability, best performing VIs (as indicated by $R^2$) within a group are formatted in bold and least performing VIs within a group are formatted in italic.

| C-flux | VI | Whole observation period | | | WW | | | WW1 | | | WW2 | | |
|---|---|---|---|---|---|---|---|---|---|---|---|---|---|
| | | Inter-cept | slope | $R^2$ | Inter-cept | slope | $R^2$ | Inter-cept | slope | $R^2$ | Inter-cept | slope | $R^2$ |
| NEE* | EVI | 1.87 | 1.29 | 0.84 | 1.91 | 1.36 | 0.88 | 1.82 | 1.44 | 0.92 | 1.91 | 1.4 | **0.92** |
| | EVI2 | 1.85 | 1.49 | **0.87** | 1.89 | 1.56 | **0.91** | 1.78 | 1.64 | **0.96** | 1.86 | 1.71 | 0.86 |
| | NDVI | 1.67 | 1.4 | 0.86 | 1.56 | 1.63 | 0.88 | 1.56 | 1.61 | 0.9 | 1.52 | 1.72 | 0.82 |
| | GNDVI | 1.3 | 2.06 | 0.85 | 1.17 | 2.28 | 0.88 | 1.17 | 2.24 | 0.94 | 1.05 | 2.51 | 0.8 |
| | SAVI | 1.79 | 1.7 | **0.87** | 1.8 | 1.82 | **0.91** | 1.7ds | 1.91 | 0.95 | 1.79 | 1.93 | 0.86 |
| | NDWI | 2.26 | 1.04 | 0.76 | 2.35 | 1.09 | 0.77 | 2.14 | 1.57 | **0.96** | 2.39 | 0.92 | *0.62* |
| | S2REP | -33.61 | 0.05 | 0.76 | -33.8 | 0.05 | 0.81 | -37 | 0.05 | 0.92 | -31.54 | 0.05 | 0.66 |
| | SR | 2.15 | 0.05 | *0.65* | 2.27 | 0.04 | *0.68* | 2.26 | 0.04 | *0.77* | 1.98 | 0.11 | **0.92** |
| | *mean* | - | - | 0.81 | - | - | 0.84 | - | - | 0.92 | - | - | 0.81 |
| GPP* | EVI | 2.28 | 1.24 | 0.77 | 2.15 | 1.52 | 0.83 | 2.28 | 1.38 | 0.8 | 2.12 | 1.56 | 0.78 |
| | EVI2 | 2.26 | 1.41 | **0.78** | 2.13 | 1.75 | **0.84** | 2.27 | 1.54 | 0.8 | 2.06 | 1.89 | **0.81** |
| | NDVI | 2.16 | 1.22 | 0.65 | 1.87 | 1.65 | *0.67* | 2.16 | 1.36** | *0.61**$ | 1.8 | 1.69 | 0.61 |
| | GNDVI | 1.83 | 1.8 | 0.64 | 1.45 | 2.34 | 0.69 | 1.79 | 1.95 | 0.68 | 1.34 | 2.46 | 0.6 |
| | SAVI | 2.21 | 1.6 | 0.77 | 2.04 | 2.01 | 0.82 | 2.21 | 1.76 | 0.77 | 1.98 | 2.12 | 0.8 |
| | NDWI | 2.63 | 1.04 | 0.76 | 2.63 | 1.3 | 0.81 | 2.59 | 1.48 | 0.82 | 2.63 | 1.14 | 0.74 |
| | S2REP | -32.88 | 0.05 | 0.64 | -38.1 | 0.06 | 0.77 | -38 | 0.06 | **0.94** | -32.47 | 0.05 | *0.54* |
| | SR | 2.53 | 0.05 | *0.61* | 2.56 | 0.04 | 0.62 | 2.71 | 0.03** | 0.62** | 2.22 | 0.12 | 0.76 |
| | *mean* | - | - | 0.7 | - | - | 0.76 | - | - | 0.76 | - | - | 0.71 |

Linear regressions explained on average 81% of the variability in observed daily NEE values for the whole observation period, ranging from 65% for SR to 87% for EVI2 and SAVI. Linear regressions for the two WW growing periods showed higher

explanatory power, with an average $R^2$-value of 0.84 and again, EVI2 and SAVI had the highest explanatory power as for the whole observation period with a $R^2$-value of 0.91. For the first WW growing period only (WW1), linear regressions explained on average 92%, ranging from 77% for SR to 96% for EVI2 and NDWI. For the second WW growing period (WW2), linear

regressions explained on average 81%, ranging from 62% for NDWI to 92% for EVI and SR. However, average values of intercept, slope and $R^2$ were not significantly different among evaluation periods, except $R^2$ between the whole observation period and WW1, and between WW1 and WW2. Overall, EVI2 seemed to be the VI which explained most robustly the variability in observed daily NEE values across different crops and growing seasons, however different VIs performed differently with different crops or growing conditions.

For GPP, interception and slope parameters and the coefficient of determination were highly significant (p<0.001) except slope and $R^2$ of NDVI and SR in the WW1 growing period (p<0.01). Among evaluation periods, mean values of interception, slope and $R^2$ were not significantly different. Linear regressions explained (significantly) less than for NEE, about 10% (p<0.05) for the whole observation period, 8% for the two WW growing periods, 16% (p<0.05) for WW1 and 10% for WW2. None of the mean of the regression parameters were significantly different between NEE and GPP within one evaluation period. This

suggested that GPP could be estimated with one generic regression model which is valid for different winter crops and different growing periods. However, this small data set might not allow for this general conclusion.

Like for NEE, EVI2 seemed to explain most of the variability in the GPP data for the whole observation period, the two WW growing periods, and the second WW growing period. S2REP showed the highest coefficient of determination for the WW1 period of 0.94 while being rather low for the whole observation period with a value of 0.64.

Juszczak et al. (2018) found a $R^2$-value for the linear regression between ground-based spectroscopy based NDVI and SAVI respectively, and GPP of WW of 0.56 and 0.59 (p<0.0001) respectively. Madugundu et al. (2017) achieved $R^2$-values of 0.81 (p=0.04), 0.86 (p=0.02) and 0.76 (p=0.33) for the linear regressions between GPP measurements along an irrigated maize growing season and Landsat-8 derived NDVI, EVI, and LSWI, respectively. These values are based on only 11 days of measurement which might be the reason for not showing high statistical significance. Huang et al. (2019b) reported $R^2$-values

of 0.27 and 0.69 for NDVI and BRDF (bidirectional reflectance distribution function)-corrected NDVI, 0.83 and 0.03 for EVI and BRDF-corrected EVI, and 0.81 and 0.56 for EVI2 and BRDF-corrected EVI2 for annual GPP of 5 cropland sites in the

US (rainfed maize) and Finnland (spring barley on peat). When accounting for temperature and moisture stress and photosynthetically active radiation, these values generally increased from between 0.4 and 0.6 to between 0.6 and 0.8 at the monthly scale. Overall, these results indicate a very variable performance of linear regressions between VIs and cropland GPP,

but EVI and EVI2 seam to outperform other VIs as proxies for GPP as in our case and Huang et al. (2019b).

Overall, our regressions for daily NEE and GPP showed a better fit to the C flux data than a similar study for grassland (Noumonvi et al., 2019). This might be attributable to the matched source area of EC and satellite data here but more importantly, managed crops have a very distinct growing cycle which might also explain the better regression results. Still, our GPP regression model of NDVI and SAVI explained about 10% more than the GPP regression models of Juszczak et al.

(2018). The study by Madugundu et al. (2017) might not be considered robust enough for comparison for the hardly significant $R^2$-values based on only 11 data pairs.

Further, our data set allowed to conclude that different crops could benefit from linear regression models based on different VIs for NEE but this needs to be verified with more data.

### 3.5 Evaluation of estimated daily C fluxes

Continuous daily C fluxes were estimated by applying the linear regression models (Table 4) to daily interpolated VI values. The evaluation is discussed along three aspects: (1) statistical measures of association and coincidence between estimates and measurements, for the different VIs, among observation periods, and between NEE and GPP, (2) comparing statistical measures for the temporal transferability (regressions of WW1 to estimate WW2 C fluxes) and (3) absolute errors in terms of the amount of seasonal accumulated C of NEE and GPP for WW2, from linear regressions based on WW1, to results from

dedicated crop-ecosystem models simulating WW and satellite-data-model-fusion approaches estimating cropland C fluxes.

### 3.5.1 Association and coincidence

Overall, the order of statistical performance among VIs (Table 5) changed slightly compared to the evaluation of the linear regressions (Table 4).

Estimates of daily NEE values showed a better fit to the measured data than GPP estimates. However, only mean correlations

(ρ) and RMSEs were systematically statistically different between NEE and GPP when comparing within observation periods

(Welch Two Sample t-test, R). Individual statistical measures did not vary systematically, i.e. showing consistent improvements or downgrades, among evaluation periods (Table 5).

The mean coincidence of NEE estimates in terms of $R^2$, RMSE and E improved from the whole observation period over WW to WW2, however, this improvement was only significant for mean $R^2$- and RMSE-values and only from the whole observation period to WW2. This supported the hypothesis that different VI models are needed to estimate C fluxes of individual crops. For the whole observation period, NEE was best estimated by EVI2, GNDVI and SAVI with an average RMSE of 2.13 g C $m^{-2}$ $d^{-1}$ and a modelling efficiency of 0.73. For the two WW growing periods, estimates with S2REP produced the lowest RMSE and the best modelling efficiency of 1.72 g C $m^{-2}$ $d^{-1}$ and 0.81, respectively. The lowest performance was achieved with SR for all observation periods, except for $R^2$ for WW2. For WW2, GNDVI gave the lowest RMSE of 1.35 g C $m^{-2}$ $d^{-1}$, the second highest $R^2$-value of 0.87 and a modelling efficiency of 0.86. Wohlfahrt et al. (2010) modelled daily NEE values of two grassland sites for one and two years, respectively, using a light response model for GPP and a simple linear relationship between measured air temperature and measured Reco. The parameter estimates of maximum GPP and apparent quantum yield of the light response model were based on the linear relationship to NDVI. This model explained 60% and 80% of measured daily NEE at two different grassland sites, respectively. However, NDVI values were derived from ground based light sensors rather than satellites thus exhibiting a different spatial resolution and representativeness than the satellite-derived VIs of our approach. Furthermore, the model was driven by half-hourly photosynthetically active radiation measurements and NEE values were simulated for the same years which were used for their model parameterisation. Still, although using half-hourly radiation measurements to simulate daily NEE, the fraction of data variability explained was similar or less compared to our results for NDVI.

**Table 5** Statistical evaluation of the final estimation of daily NEE and GPP from interpolated daily VI values for the whole observation period from the first to the last satellite image, i.e. 22 Mar 2020 to 16 Aug 2022, for the two WW growing periods (WW, from sowing to harvest each) and the statistical evaluation of the estimation of the NEE and GPP fluxes for the second WW growing period (WW2) using the linear model from the first WW growing period (Table 4). All ρ- and $R^2$-values were statistically significant at the level of $p < 0.001$ ('***'). To increase readability, best performing VIs (as indicated by $R^2$) within a group are formatted in bold and least performing VIs within a group are formatted in italic.

| C-flux | VI | ρ | $R^2$ | RMSE | E | ρ | $R^2$ | RMSE E | E | ρ | $R^2$ | RMSE | E |
|---|---|---|---|---|---|---|---|---|---|---|---|---|---|
| | | whole observation period | | | | WW | | | | WW2 | | | |
| NEE | EVI | 0.86 | 0.73 | 2.25 | 0.7 | 0.89 | **0.89** | 1.82 | 0.76 | 0.79 | 0.84 | 1.52 | 0.83 |
| | EVI2 | **0.87** | **0.75** | **2.15** | **0.73** | **0.9** | 0.83 | 1.89 | 0.77 | 0.8 | 0.87 | 1.45 | 0.84 |
| | NDVI | 0.8 | 0.75 | 2.23 | 0.7 | **0.9** | 0.78 | 2.11 | 0.72 | 0.83 | 0.86 | 1.36 | 0.86 |
| | GNDVI | **0.88** | **0.77** | **2.13** | **0.73** | **0.9** | 0.8 | 1.98 | 0.75 | **0.86** | **0.87** | **1.35** | **0.86** |
| | SAVI | **0.87** | **0.76** | **2.12** | **0.73** | **0.9** | 0.83 | 1.88 | 0.78 | 0.81 | 0.88 | 1.38 | 0.86 |
| | S2REP | 0.84 | 0.74 | 2.39 | 0.66 | 0.86 | 0.82 | **1.72** | **0.81** | 0.88 | 0.82 | 1.73 | 0.77 |
| | SR | 0.87 | 0.53 | 3 | *0.47* | 0.89 | 0.59 | 2.75 | *0.52* | 0.81 | **0.89** | 2.72 | *0.44* |
| | *mean* | 0.86 | 0.72 | 2.32 | 0.67 | 0.89 | 0.79 | 2.02 | 0.73 | 0.83 | 0.86 | 1.64 | 0.78 |
| GPP | EVI | 0.76 | 0.76 | 3.32 | 0.71 | 0.74 | 0.8 | 3.24 | 0.74 | 0.68 | 0.85 | 2.8 | 0.72 |
| | EVI2 | **0.76** | **0.76** | **3.26** | **0.72** | 0.73 | 0.8 | 3.25 | 0.74 | 0.69 | **0.88** | **2.6** | **0.75** |
| | NDVI | 0.68 | 0.63 | 4.1 | 0.57 | 0.66 | 0.65 | 4.05 | 0.6 | 0.67 | 0.77 | 4.17 | 0.38 |
| | GNDVI | 0.7 | 0.64 | 3.96 | 0.58 | 0.69 | 0.69 | 3.75 | 0.66 | 0.7 | 0.78 | 3.73 | 0.5 |
| | SAVI | 0.75 | 0.75 | 3.34 | 0.71 | 0.73 | 0.78 | 3.32 | 0.73 | 0.69 | 0.87 | 2.75 | 0.73 |
| | S2REP | **0.79** | 0.73 | 3.33 | 0.71 | **0.83** | **0.85** | **2.48** | **0.85** | **0.74** | 0.81 | 2.72 | 0.74 |
| | SR | 0.69 | 0.53 | 4.53 | *0.46* | 0.69 | 0.55 | 4.64 | *0.47* | 0.65 | 0.84 | 5.05 | *0.09* |
| | *mean* | 0.73 | 0.69 | 3.69 | 0.64 | 0.72 | 0.73 | 3.53 | 0.68 | 0.69 | 0.83 | 3.4 | 0.56 |

Individual statistical evaluation measures for GPP estimates did not differ significantly among evaluation periods, except ρ and $R^2$ between the whole observation period and WW2. Here, only $R^2$ increased from the whole observation period to WW2.

And as for NEE, the individual performance of VIs varied among evaluation periods. For the whole observation period, EVI, EVI2 and SAVI had a very similar performance for all statistical measures and only the correlation (ρ) of S2REP showed the highest values of 0.79 for this evaluation period.

### 3.5.2 Evaluation of temporal transferability by comparing statistical measures

The mean coincidence of our NEE estimates for WW2 in terms of $R^2$ and RMSE was 0.86 and 1.64 g C $m^{-2}$ $d^{-1}$ respectively (Table 5). These wer very similar to simulation results from a dedicated crop model (SAFY-CO$_2$) driven by satellite data for WW with $R^2$-values of 0.78-0.9 and RMSE-values of 1.09-1.59 g C $m^{-2}$ $d^{-1}$ (Pique et al., 2020). Pique et al. (2020) simulated eight cropping years of WW at two agricultural sites in southwest France with observed annual NEE values ranging from -208 +/-19 to -410 +/-45 g C $m^2$ $yr^1$. Arora (2003) simulated NEE of one WW growing season (doy 51-151) at an agricultural site

in north-central Oklahoma (US) using a coupled land surface terrestrial ecosystem model. RMSE-values of 2.37 and 2.69 g C $m^{-2}$ $d^{-1}$ and $R^2$-values of 0.7 and 0.76 were achieved (Arora, 2003) which indicated a slightly lower accuracy than our approach. The ORCHIDEE-STICS (dynamic global vegetation model with the process-oriented crop model STICS) and the SPA (soil plant atmosphere) models showed lower mean $R^2$-values of 0.75 and 0.83 and similar to higher RMSE values of between 1.2 and 3.1 and 1.47 g C $m^{-2}$ $d^{-1}$, respectively, (Sus et al., 2010; Vuichard et al., 2016) compared to our values. ORCHIDEE-STICS

and the SPA model were used to simulate seven WW growing seasons at seven agricultural sites and eight WW growing seasons at six agricultural sites in central Europe, respectively (Vuichard et al., 2016) (Sus et al., 2010; Vuichard et al., 2016). The simulations were carried out at mostly the same European sites where observed NEE values ranged from -529 to -169 g C $m^2$ (Sus et al., 2010) and from -451 to 15 g C $m^2$ (Vuichard et al, 2016).

A comprehensive crop model inter-comparison study, carried out at the already mentioned European cropland sites, however, showed a very high variability in model performances simulating C fluxes of WW. τ-values (Kendall correlation coefficient; as used in Wattenbach et al., 2010) ranged from 0.28 to 0.81 (mean=0.58) for growing season NEE and modelling efficiency (E) values spread between 0.31 and 0.87 (mean=0.55) (Wattenbach et al., 2010) while our respective mean E-values ranged from 0.44 to 0.86.Our GPP estimates showed lower accuracy than the modelling exercise that was previously mentioned in

which $R^2$ and RMSE values were 0.86-0.96 and 0.9-2.79 g C $m^{-2}$ $d^{-1}$ for GPP (Pique et al., 2020) as compared to our mean

values of 0.83 and 3.4 g C m⁻² d⁻¹ for WW2, respectively. GPP was simulated similarly well with the models in Wattenbach et al. (2010) as with our approach. Our mean ρ for GPP estimates for WW2 was 0.69 and mean E was 0.56, the respective mean values from Wattenbach et al. (2010) were 0.58 for τ and 0.65 for E. ORCHIDEE-STICS showed R values of above 0.7 for growing season GPP of WW (Vuichard et al., 2016).

The differences between observed and estimated NEE and GPP of this study are visually exemplified for EVI2 in Figure 5.

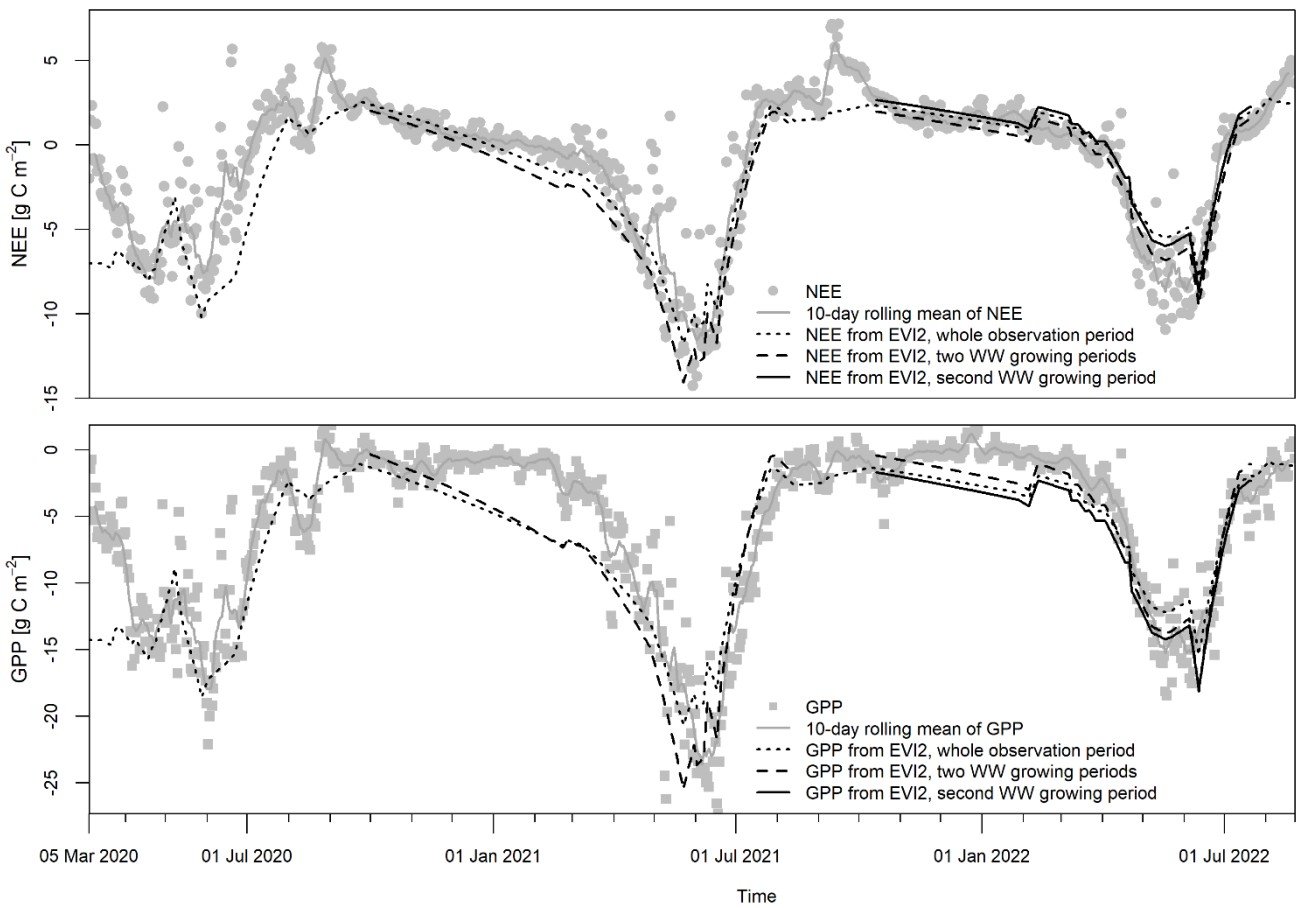

**Figure 5** Measured (grey points and lines) and estimated daily NEE and GPP for the whole period (black dotted), for the two WW growing periods (black dashed) and for the second WW growing period (black solid). Estimated NEE and GPP values

for the whole period were imputed from the linear regression from the whole observation period, NEE and GPP values for

the two WW growing periods were calculated from the linear regression based on the two WW growing periods and, NEE and GPP of the second WW growing period were based on the linear regression from on the first WW growing season only.

We also put our results into perspective of a number of studies which use high resolution satellite based VIs in combination with mechanistic or machine learning approaches to estimate daily C fluxes for croplands. Fu et al. (2014) diagnosed cropland NEE using a regression tree model in combination with highly processed Landsat imagery and NDVI and EVI. At the four cropland sites (rainfed maize, US) in their study, they achieved $R^2$ and RMSE values of 0.66 to 0.91 and 1.95 to 2.34 g C m$^{-2}$ d$^{-1}$, respectively, where observed and diagnosed NEE were -0.68 (0.24 standard deviation) and -1.4 (0.28 standard deviation) g C m$^{-2}$ d$^{-1}$, respectively. They speculated that the overestimation of C uptake is due to the "ill parameterization" of their statistical model because of the low number of cropland sites. They do not leverage the information about the different site characteristics to further explore the large variability of model performance between the different sites. Bazzi et al. (2024) evaluated the performance of a modified vegetation-photosynthesis-respiration model supported by Sentinel-2 LSWI and EVI, simulating NEE, GPP and Reco at 12 European cropland sites (range of crops including WW) for the years 2018-2020. $R^2$-values ranged from 0.72 to 0.86 for daily NEE, 0.81 to 0.86 for daily GPP and 0.38 to 0.77 for daily Reco. Note however, that this accuracy measure was no cross-validation result but for the same sites-years as used for parameter optimisation. A combination of mechanistic modelling, Sentinel-2 or Landsat-8 optical remote sensing data and machine learning algorithms was actually needed to achieved $R^2$-values of 0.91 and 0.82 with Sentinel-2 and Landsat-8 data, respectively, for daily GPP estimates for 6 cropland site years (soybean, winter wheat, winter barley, colver) (Wolanin et al., 2019).

While all these approaches do generally show similar or higher accuracy as our approach, a much higher effort is needed to derive NEE and GPP estimates.

### 3.5.3 Absolute errors of temporal transferability

Absolute deviations between the measured seasonal C fluxes of NEE and GPP and our estimates for every VI is displayed in Table 6. Total NEE estimates ranged from an underestimation of C uptake of 195.8 g C m$^{-2}$ (85.66%, S2REP) to an overestimation of 57.24 g C m$^{-2}$ (25.04%, SR) (Table 6). Best estimates were achieved with NDVI and GNDVI with an

overestimation of C uptake of 33.36 g C m$^{-2}$ (14.59%) and an underestimation of 40.98 g C m$^{-2}$ (17.93%), respectively. Both

VIs had the lowest RMSE-values of 1.36 and 1.35 g C m$^{-2}$ respectively when estimating daily C fluxes (Table 5).

The model ensemble of Wattenbach et al. (2010) exhibited simulation errors of total annual C sums ranging from an overestimation from the measured C uptake of 204 g C m$^{-2}$ to an underestimation of C uptake of 217 g C m$^{-2}$ across the 5 WW site-years and 3 crop models used in their study. The simulated seasonal C uptake (NEE) by ORCHIDEE-STICS ranged from an overestimation of C uptake of 251 g C m$^{-2}$ (1673%) to an underestimation of C uptake of 321 g C m$^{-2}$ (108%) (Vuichard et

al., 2016). The SPA model had a tendency to overestimate C uptake (NEE) ranging from 5 (0.9%) to 289 g C m$^{-2}$ (92%) at 8 WW site-years in Europe (Sus et al., 2010). This comparison highlights again that sophisticated mechanistic ecosystem models are not a priori superior to simple regression approaches.

**Table 6**: Observed and estimated and the respective difference of cumulated C fluxes for the second WW growing period (WW2) using WW1-based regression models. For the interpretation of NEE results: negative absolute values and positive % values denote an underestimation of C uptake compared to the observed C uptake. For the interpretation of GPP results: negative absolute values and negative % values denote an overestimation of C uptake compared to the observed C uptake.

| C-flux (measured) | VI | Estimates [g C m$^{-2}$] | Total difference [g C m$^{-2}$] | Difference [%] |
|---|---|---|---|---|
| | | WW2 | | |
| NEE -228.58 | EVI | -117.00 | -111.58 | 48.81 |
| | EVI2 | -62.48 | -166.10 | 72.67 |
| | NDVI | -261.93 | **33.36** | **-14.59** |
| | GNDVI | -187.60 | **-40.98** | **17.93** |
| | SAVI | -86.38 | -142.19 | 62.21 |
| | S2REP | -32.78 | -195.80 | 85.66 |
| | SR | -285.81 | 57.24 | -25.04 |
| | *Mean* | -147.71 | -80.86 | 35.38 |
| GPP 1165 | EVI | 1712.15 | -547.18 | -46.97 |
| | EVI2 | 1663.08 | -498.12 | -42.76 |
| | NDVI | 2090.09 | -925.12 | -79.41 |
| | GNDVI | 1945.69 | -780.73 | -67.02 |
| | SAVI | 1723.97 | -559.01 | -47.98 |
| | S2REP | 1458.13 | **-293.17** | **-25.17** |
| | SR | 2018.80 | -853.84 | -73.29 |
| | *Mean* | 1801.7 | -636.74 | -54.66 |

Discrepancies in our best NEE estimations were in the same range as our estimated uncertainty of 0.8 and 13 g C m$^{-2}$ for 2020/2021 and 2021/2022, respectively, and reported NEE uncertainties of 40 g C m$^{-2}$ of WW crops (Aubinet et al., 2009; Béziat et al., 2009). Further, estimation-uncertainty is smaller than the difference of total growing season NEE between the two WW growing periods 2020/2021 and 2021/2022, respectively, which was 311 g C m$^{-2}$ (i.e. 3.11 t C ha$^{-1}$, Table 3).

Larger differences occurred between measured and estimated GPP fluxes. Estimated GPP overestimated absolute measured flux values from 25% up to 79%. The latter was due to the overestimation of C uptake during winter (as exemplified in Figure 5) when VI values indicated a vital crop growth inferred from relatively high crop greenness values as explained earlier (see section 3.2). Here, the previously-mentioned issue of diverting NEE and VI signals after sowing and during winter (section 3.2) showed its effect, causing a larger discrepancy between observed and estimated GPP for the WW growing period. Deviations of our estimated GPP values were higher than simulated GPP values of the mechanistic crop model ORCHIDEE-STICS, ranging from an underestimation of 42% to an overestimation of 20% for WW growing seasons (Vuichard et al., 2016). A variety of explanations are discussed for the ecosystem models failing to simulate C exchange over growing seasons. Models face problems to represent phases of low C fluxes (such as winter) (Dietiker et al., 2010; Wattenbach et al., 2010) or the models assume post-harvest phases as bare soil ignoring regrowth from weeds and/or leftover seeds and thus underestimating C uptake (Lu et al., 2017; Vuichard et al., 2016). The importance of capturing spontaneous re-growth which can usually not be simulated by mechanistic crop-growth models unless specifically parameterized for has been pointed out in relation to the advantages of using remote sensing data in crop modelling. Especially the knowledge of key dates, such as time of emergence, maximum vegetation, start of senescence and harvest, determine the accuracy of model estimations (Pique et al., 2020). While our approach struggled with the low-flux time during winter as well, the high resolution of Sentinel-2 imagery, especially during non-winter seasons, is well suited to pick up the plant growth dynamics at the respective key dates without knowing them explicitly. In conjunction with the good performance of our estimates using linear regression only, our approach constitutes a promising alternative of very low data demand to estimate C exchange of a highly dynamic and heterogeneous, small parceled landscape.

Vuichard et al. (2016) and Sus et al. (2010) further discussed issues related to the representation of phenology in the models which is a determining factor for the subsequent calculation of C fluxes. Now, using the actual spectral optical properties of crops directly - as in our approach - might constitute an advantage tracking actual phenology and thus the associated evolution of C fluxes.

### 3.6 Strengths and weaknesses of the approach

Finally, the strengths and weaknesses of the approach are briefly discussed. Weaknesses were the use of simple linear regression models, which are empirical and not mechanistical, and, a lot of evaluation and proof of concept is still needed before the approach can be applied spatially.

As explained in the introduction, NEE is only indirectly linked to spectral VIs via the direct correlation of GPP with spectral VIs and the observed correlation between GPP and Reco. Since the sum of negative GPP and positive Reco gives NEE the link of NEE to spectral VI directly was justified. In our study, GPP and Reco significantly ($p<0.001$) correlated by -0.63 (spearman's) but GPP and NEE correlated even better by 0.95 ($p<0.001$) (using cases with NEE qc=0 measurements only). Furthermore, correlations between NEE and VIs were stronger than correlations between GPP and VIs (Figure 4). Considering these highly significant and strong correlations, a direct empirical and linear link between GPP and VIs and even more between NEE and VIs seemed justified and sufficiently proven.

However, the high correlation between NEE and VI hides the problematic diversion of signals during winter. Here, the very few VI images during winter caused a better correlation than might have been observed with more winter VI data. Thus, the decoupling of the signals needs to be further addressed. Theoretically, lowest VI values should be linked with bare soil or very little vegetation and thus mainly Reco. This assumption is invalidated by the winter increase of VI values. If this non-active-growing period was treated differently than the spring-summer period it could be cut out from the correlations and would need to be replaced by another assumption, such as assuming a baseline winter C flux. The associated questions would be how variable baseline respiration at croplands is, what the proportion of winter fluxes to the total is, and what impact that would have on the total results if it varies.

The most prominent question which now remains is if the linear regressions fitted here hold for other crops than winter grains, such as maize or root crops, or are they grain-crop type specific? Juszczak et al. (2018) argues that a generic single relationship between VI and C flux can be valid for a range of different crops.

The strengths of this approach are the low data demand, the straightforwardness and the accuracy compared to ecosystem models and satellite-data-model-fusion approaches, respectively, which are the most sophisticated approaches to estimate

spatial C exchange, and, by monitoring plant "greenness" directly, the highly complex plant growth is integrated into a representative signal.

## 4 Conclusions and outlook

The observed $CO_2$ dynamics of the presented cropland site were representative for a typical winter rape and winter wheat cropland in Europe. The site was thus suitable for developing a generic approach of linking remote sensing data with EC measurements. A linking approach consisting of multiple evaluation steps was developed by appropriately accounting for spatial alignments between EC measurement footprints and remote sensing data of fine spatial resolutions. This rigorous linking approach was applied to a range of VIs to evaluate their strengths and weaknesses in estimating daily $CO_2$ fluxes for the search of the most promising VIs. The general validity of the approach was shown by the high and statistically significant linear correlations between C fluxes and VIs. However, the ranking of the suitability of VIs differed amongst the evaluation by the linear correlation, the linear regression, and the temporal transferability, indicating no single stable or universally superior VI for daily $CO_2$ flux estimation. While linear regressions suggested S2REP as the most promising VI to estimate WW NEE, NDVI and GNDVI were the best for the temporal transferability of WW NEE estimations. Overall, the approach leads to results similar to the results of complex ecosystem models or sophisticated satellite-data-model-fusion approaches which justifies the data-driven and data-lean approach. Relatively small estimation errors at this stage of research further suggest that this approach is a promising method for tracking C exchange remotely over croplands. Future work should address mainly three questions: does one generic relationship between VIs and C fluxes hold for other crop types and/or climate conditions as well? Which VIs are most suitable for estimating which C flux? And, does any additional information, such as temperature or radiation for light-use-efficiency modelling, further improve accuracy? Or from a more overarching perspective: which processes or data uncertainties explain the gap between measured and estimated C fluxes?

EC data quality control

To assure a robust time series of half-hourly flux measurements, "qc0 data" were further screened for outliers by calculating half-hourly median and standard deviations of a 30-day moving window and testing each half-hour flux measurement against the respective half-hour statistics. Fluxes exceeding the median +/-2 (daytime) and +/-3 (nighttime) standard deviations were excluded from the time series (similar to Goodrich et al. (2015)). This was followed by a visual inspection of plotting diurnal

half-hourly fluxes against monthly diurnal means. This yielded some extreme values which were within the previously defined bounds but which still strongly biased the median and standard calculation in the previous step in times of high occurrence of gaps. These values were removed and the 30-day moving window statistics loop was re-iterated.

**Appendix B**

Filtering "main field" fluxes with FP modelling

Figure 1 (in the main text) shows the cumulative source area of the EC measurements extending over adjacent fields. To construct a spatially representative NEE time series of the "main field" excluding surrounding areas, measured fluxes were combined with FP modelling following the approach of Göckede et al. (2004). The source area of the flux measurements is heterogeneous in space (Figure 1) and time and its distribution varies with changing meteorological conditions. Stable atmospheric conditions enlarge the FP, usually during night time, while unstable conditions during daytime decrease the size

of the FP. Thus, each flux measurement carries a mixed signal of fluxes originating to a variable degree from different surface areas. To quantify the contribution of different surface areas to the total half-hourly flux analytical FP models are employed (Göckede et al., 2004). Here the analytical FP model by Kormann and Meixner (2001) was used. The land use map was constructed by classifying visible homogeneous land use areas from a Google Earth Image (same as in Figure 1 of the main text) in form of a discrete matrix (Figure B1).


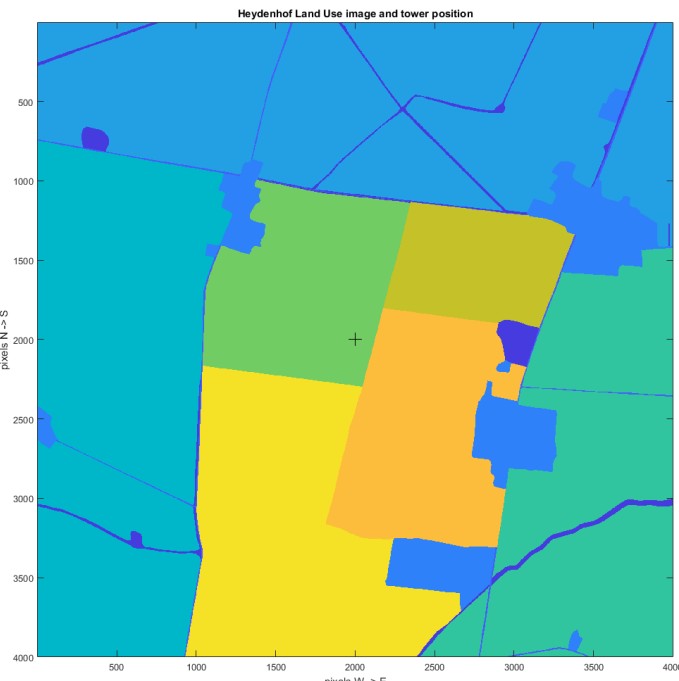

**Figure B1** Land use image of Heydenhof used for applying the analytical FP model. Different colors distinguish the different fields and land use types used to determine source area contributions. The "+" indicates the location of the EC tower. The green field in the middle where the "+" symbol is located is the "main field".


Gaps in wind direction measurements caused respective gaps in the FP results. The 0.6% of missing wind direction measurements, with the longest gap of 9 hours, were filled by linear interpolation. In turn, the gaps in the FP results were filled by assigning the gaps the average values of FP results of respective 1° wind direction bins.

According to Göckede et al. (2008) fluxes with a 95% contribution of a specified source area to the total flux are termed 665 "homogeneous measurements" while fluxes with an 80-95% contribution are still regarded as "representative measurements". However, these limits were set based on intensive pre-analyses and practicability.

**Appendix C**

Gap-filling and partitioning of flux data

For calculating C budgets from NEE data, a continuous time series is required. To avoid periods of insufficient turbulence which violate EC assumptions and could bias nighttime fluxes, i.e. ecosystem respiration, data were filtered by a u*-threshold that determines low turbulence conditions. Here, the u*-threshold was calculated by the moving point method (Papale et al., 2006). Subsequently, the NEE time series was gap-filled by the marginal distribution sampling (MDS) approach of Reichstein et al. (2005) which has widely been used for arable EC flux measurements (Béziat et al., 2009; Pastorello et al., 2020).

U*-estimation, gap-filling, uncertainty estimation by boot-strapping and flux partitioning was carried out with the R-package "REddyProc" (Wutzler et al., 2018), available from https://www.bgc-jena.mpg.de/bgi/index.php/Services/REddyProcWebRPackage.

**Appendix D**

A comprehensive cropland soil C budget encompasses a number of C flows in addition to GPP, Reco, manure and seed inputs 680 and harvest exports. These include C losses due to fire, wind and water erosion, leaching of dissolved organic C (DOC) and volatile organic compound (VOC) losses (Ciais et al., 2010), exchange in form of CO and $CH_4$ and C input from deposition (Waldo et al., 2016). Losses due to fire can be ignored for our field. Erosion and deposition can be assumed to cancel out due to the surrounding area being of the same nature as our main field. CO, $CH_4$ and VOC can be considered negligible for a regular cropping field (Waldo et al., 2016) as well as leaching losses of DOC (Siemens et al., 2012).


**Code availability**

The MATLAB, R and JavaScript codes for flux and satellite data processing including quality control, analyzes and visualization as produced for this paper is available via Gottschalk et al. (2024a).

**Data availability**

All flux and metadata are openly available via the European Fluxes Database (http://www.europe-fluxdata.eu/home). The version of half-hourly flux data and auxiliary meteorological data for this article in standard EddyPro output, a shape file outlining the main field boarders, and the TERENO precipitation data are available via Gottschalk et al. (2024b).

**Author contribution**

Pia Gottschalk: Conceptualization, Methodology, Software, Formal analyses, Validation, Visualization, Writing - Original
Draft, Review & Editing. Aram Kalhori: Writing – Review & Editing. Christian Wille: Data Curation, Software, Writing – Review & Editing. Zhan Li: Software, Writing – Review & Editing. Torsten Sachs: Resources, Writing – Review & Editing, Supervision, Project administration, Funding aquisition. All authors have read and agreed to the published version of the manuscript.

**Competing interests**

The authors declare that they have no conflict of interest.

**Acknowledgments**

We thank Karl Kemper (GFZ intern) for preparing the land use matrix for the footprint modelling and providing Figures 1 and
B1. We further acknowledge Youping Li who had developed the first draft of the GEE scripts during a scientific visit at GFZ.

**Financial support**

PG acknowledges funding by the German Federal Ministry of Food and Agriculture (BMEL) in the frame of the ERA-NET FACCE ERA-GAS project GHG-manage, grant No. 2817ERA10C. FACCE ERA-GAS has received funding from the European Union's Horizon 2020 research and innovation program under grant agreement No. 696356. We further used infrastructure of the Terrestrial Environmental Observatory Network (TERENO).

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
