# Peer review of "C flux dynamics at Heydenhof"

_EGUsphere, 2023_

## Author Comment (AC1)

Overview

This manuscript combines field scale EC data and satellite derived VIs to estimate the carbon flux over a local cropland in Europe. Several kinds of VIs are used to estimate NEE and GPP based on linear regression models. Their method is reasonably straightforward, and assumption is clearly stated. The results are well investigated from multiple aspects, which makes their conclusions more concrete. Their estimated values including uncertainties are fairly compared to previous works. The authors explain the strengths and weaknesses of their approach very well. Some figures and tables may be a little hard to read/follow to readers, though.

Thank you for the benevolent and constructive evaluation of our manuscript! In the following I will reply to the non-critical comments briefly and adequately to the more critical comments.

Followings are my minor comments and suggestions.

In the abstract, please clarify the size/location of the study field.

Total polder area is around 550ha and since 2005 around 272 ha are permanently inundated. The inundated open water area is around 10 ha.

L20: Please define "RMSE"

Sure! We will add a reference for the formula used as the "root mean square error" is sometimes defined differently.

L79: "limits"

Will be corrected. Thanks!

L86: Please define "NIR"

Sure! NIR = near infrared

L114: Please define "a.s.l."

a.s.l. = above sea level. We will just write it out.

L134: Data "were" measured…

Thanks! Will be corrected.

L141: For date expression, please be consistent ("5$^{th}$ March" or "23 August").

We will revise the manuscript accordingly.

L164: Please define "SCL".

SCL = "scene classification map". Will be added.

L198: There is no "section 0" in this manuscript.

Sorry about this mistake. This is obsolete.

L222: It will help to put the total NEE, GPP and Reco for each growing season in Fig.2 as you did for the climate parameters.

Yes, very nice idea. We will add this information to Fig. 2. Thanks!

L254: "apart from" Do you mean "except for"?

Yes, thanks. Will be revised.

L277-278: The VI observations are sparse compared to NEE/GPP, so it might be hard to tell "these VIs lag about 18 days behind the NEE signal for the senescence period of WR" from the Fig. 3. Please specify how you verify it.

True, VI observations are generally sparse. "about 18 days" is only an average across VIs based on the observation that given VI values had reached a plateau at the same time as of max. C uptake (= min NEE) and do not drop at the same time as NEE decreases, as another high VI value can be observed about 18 days later before the next VI value is much lower.

L308: Please define "lowess (should be LOWESS)".

We will change to "fitted line" as explained in the manual of the R-function used: chart.correlation of the R-package "PerformanceAnalytics".

L287-288: Why the signs of correlations are opposite of yours?

Signs are opposite because Baldocchi (2008) and Baldocchi et al., (2015) use the opposite sign for GPP as we do. Which sign to report final GPP values in does not follow a convention. Both can be used and the absolute correlation, which is the important information, does not depend on the sign.

L289: This sentence is hard to understand and seems not quite matching the results shown in Fig. 4.

Agreed. It should be along the lines: "NEE correlated highly significant ($p<0.001$) with all VIs except MNDWI which showed a lower level of statistical significance ($p<0.01$). GPP correlated highly significant ($p<0.001$) with all VIs except MNDWI ($p<0.01$) and NDSVI ($p<0.05$) which both showed a lower level of statistical significance.

L321 and L324: "nir"--> please define and change to "NIR".

We will change "nir" to "NIR". The definition will be given according to a previous comment by the reviewer (L86).

L322: "swir" --> please define and change to "SWIR".

We will change "swir" to "SWIR" and give the respective definition of "short-wave infrared".

L323: "amongst the highest" is not true. Do you mean "one of the highest"?

For me as a non-native English speaker the difference is very subtle here but I am happy to change to "one of the highest"!

L383: Please define "E". "RMSE" does NOT "increased" (but improved).

Yes, "increased" is wrong. Will be changed to "improved". "E" is defined in L183.

L405: It is not very clear that $R^2$-values are calculated between what parameters.

Agreed. We will change to "Jaszczak et al. (2018) found a $R^2$-value for the linear regression between NDVI and SAVI, respectively, and GPP of WW of 0.56 and 0.59 (p<0.0001).

L422: "τ" --> Do you mean "ρ"?

It is "τ" (Kendall tau rank correlation coefficient) as this is the correlation coefficient used in Wattenbach et al. (2010). We will add a comment noting the different coefficients used.

L434-439: These sentences are hard to understand. "overestimation" and "underestimation" with respect to what? Your estimated results?

Over- and underestimations relative to the measured values at the simulated sites used in the cited publication. We will add this information to the text.

L435: "204" and "217" are for cumulative C flux, right? Please mention that and specify the time period.

Correct. These values are annual sums. We will add this to the text.

L440: "85%" --> "85.66%".

Agreed. Will be changed.

L442: Please add ", respectively" after "(17.93%)".

Sure, will be added.

L443: "1.35 and 1.36" --> "1.36 and 1.35". "(Table 6)" --> "(Table 5)".

Yes, thanks for spotting this! Will be changed.

L451: Please add ", respectively" after "2021/2022,".

Sure, will do. Thanks!

L452-453: This sentence is a little hard to understand. I do not see how it is related to Table 3. Where the number "311 g C m$^{-2}$" come from?

"311 g C m$^{-2}$" is the difference of NEE values of 2020/2021 and 2021/2022 respectively, after converting from t C ha$^{-1}$ to g C m$^{-2}$. We will change the sentence to "Further, estimationuncertainty is smaller than the difference of annual NEE values between the two WW growing periods 2020/2021 and 2021/2022, respectively, which was 311 g C m$^{-2}$ (Table 3)."

L467: Please add "," after "harvest".

Thanks, will do.

In sections 3.5.2 and 3.5.3, more detailed explanations of the models are needed such as the model simulation years (is there any inter-annual variability)? Readers might understand better if section 3.5.3 starts with explanation of Table 6, and then give comparisons with references.

Good point! To address this, we will add an additional sentence to section 2.7 introducing the reader to the comparison of our results to results of simulations studies by mechanistic crop models. In section 3.5.3 we will add the information of model simulation years and inter-annual variability from the simulation studies. Furthermore, we will be happy to start section 3.5.3 with explaining Table 6. This is exactly the comment which is needed from an independent reader to improve the readability of the manuscript. Thanks again!

Figure 1: It is a little too hard to understand what the percentage of "cumulative FP area" means. How was it calculated? It is not clear the definitions of "main field" and "area of interest".

We will write out FP as "foot print" area in the caption of Figure 1 and guide the reader to Appendix B, where the foot print modelling is explained. We will there add a sentence in which the cumulative foot print area is explained as the source are of the fluxes, and the isolines denote the cumulative contribution of that area to the flux signal over the measurement period. Further, we will remove "area of interest" as it is synonymously used for main field (Appendix B, L551) and occurs only once in the whole manuscript.

Figure 2 (The top panel): Please specify the duration of "flowering of WR". (the first 2 top panels): please explain the gray dots and black solid lines in the figure caption. Also, please differentiate the GPP and Reco by using different color or line type. I do not see the "urea spreading" in 2020/2021 listed in Table 1.

We will add a horizontal segment to show the length of WR flowering. We will obviously add an explanation of the gray dots and solid lines. Sorry for forgetting. We will adjust the line types of GPP and Reco to better differentiate between them. We will add a vertical line for the "urea spreading" in 2021!

Please give x-axis title as "Time" and do not use the numbers such as 5.3, 1.5, 1.7… which are hard to understand. You could show the dates (i.e. 05 Mar 2020) only for the start and end of the growing season or so. Also, it may be helpful for readers to put the x-tick values at the top of the figure, too.

Figure will be improved according to these comments.

Figure 3: Y-title for VIs should be "(-1)*VIs"? There is no explanation for blue dots. Again, the x-axis' thick name of "5.3, 1.6…" are hard to follow. Please give monthly tick marks and label them properly. Also, it would be easier for readers to compare the time series of

NEE/GPP and VIs if they have a common zero (horizontal) line. Please add legends for the vertical broken- and dotted- lines.

Good point to label the secondary y-axis with (-1)*VI. We will add the explanation for the blue dots and change the labeling of the x-axis. We deliberately have R optimize for the level of 0 of the two signals so the measured C values and the dots of the VIs align most closely. We will try out how it looks when fixing a common 0 line and if readability improves. We will add an explanation of the vertical broken- and dotted- lines. They denote sowing and harvest dates. Sorry for forgetting.

Figure 5: Please put an x-title and change the x-tick name format.

Yes, we will have an improved and uniform labelling of the x-axis throughout the figures!

Table 1: What is "ca."?

Ca. stands for "circa". We will revise to "approx."

Table 3: Please state the unit clearly.

All values are given in t C ha$^{-1}$ season$^{-1}$. We will clarify in the caption.

Table 4: Please explain what the bold and italic numbers indicate either in the caption or main text. Why are $\rho$-values not shown here (just wonder) as in Table 5? It could be better to make the line thicker between different growing periods (in Table 5, also).

Yes, we will explain bold and italic writing. Bold indicate best performing VIs within a group and italic indicate worst performing VIs within a group. We explain in the caption that all values are significant to the level of 0.001 except where indicated with '**'. Thus, we avoid printing '***' in each cell. That holds true for tables 4 and 5. We will improve readability by thickening the line between the groups.

Table 5: Please explain what the bold numbers indicate.

Bold indicate best performing measures within a group. We will explain this.

Table 6: the top of the 3$^{rd}$ column should be "NEE and GPP estimated [g C m-2]". You could add "*mean*" to Table 4 and 5 as well because it was mentioned in the main text. What does the "**" indicate? Again, please explain about the bolded numbers (it seems they are mentioned in the next, but not all of them are bolded).

Thanks for spotting the wrong labeling of column 3. Will be revised. We will add rows for mean values in table 4 and 5. '**' actually has no meaning and will be removed.

S1 The third paragraph: "Reco … but showed no response the grubbing events". Is this true? It seems the Reco increased between the first and second grubbing events. The last (fifth) paragraph, "GPP and Reco were both lower …": because the signs of GPP and Reco are opposite, "lower" is a little confusing. Could be better to say "smaller".

We will check in more detail on the response of Reco to grubbing events and revise accordingly. We will change "lower" to "smaller". Thanks.

---

## Author Comment (AC3)

Gottschalk et al. present a high-resolution investigation of the vegetation indices against the carbon fluxes from a tower in the cropland. They processed the flux observations thoroughly to ensure the data quality and estimated the fluxes based on linear models using VIs, which were then compared with crop models. They showed that observed fluxes correlate well with the field management actions, phenological development of crops, and some Sentinel-2-derived VIs (especially for GPP). They concluded with discussions of the limitations of their analyses (e.g., adopting a simple linear model).

Major comment:

In general, the data processing is thorough, and the results are well presented. Leveraging the high-resolution feature from Sentinel-2 data, their EC-based carbon fluxes and Vis are all matched to the plot scale, which addressed the common spatial mismatch issue in prior studies.

Despite the promising flux-VI relationship presented, my major concern lies in the novelty of this study and the representativeness of such flux-VI relationships across crop traits and climates (despite that the authors acknowledged till the very end). The authors suggested that their analyses indicate "the suitability and developability of the proposed approach to monitor cropland C exchange with satellite derived Vis*"* (L25-26) and acknowledged the current challenges in flux estimates being the low number of EC towers in croplands (L79-81). When considering croplands over different climates/regions, how robust is the use of a linear model to predict carbon fluxes? Would involving additional meteorological variables be more helpful (which is relative to statements on L490-491)?

Concerning the novelty of the study: many prior studies have indicated the potential of satellite images for carbon flux estimation (Sims et al. 2006, Huang et al. 2019b) but mostly at coarse spatial resolutions of MODIS-like 1km. Some recent studies explore the potential of finer-resolution satellite images such as Landsat or Sentinel-2 (Fu et al., 2014, Madugundu et al., 2017, Chen et al., 2010, Wolanin et al., 2019, Bazzi et al., 2024) but mostly only investigate one or two spectral indices while focusing on GPP over vegetated lands in general rather than dedicated cropland nor NEE. While EC is the most defensible approach to measuring ecosystem-scale carbon fluxes, many prior studies do not appropriately account for the spatial mismatch between EC measurement footprints and satellite pixel footprints. Tramontana et al. 2015 & 2016 also highlighted the need to address such a spatial mismatch. A lack in our understanding of satellite imageries' capabilities for cropland carbon flux estimation and monitoring is a rigorous evaluation of comprehensive spectral indices from finer-resolution satellite imageries by appropriately leveraging EC-based carbon flux measurements. Our study fills in this knowledge gap through a detailed and rigorous analysis into the relationship between a large list of satellite-imagery VIs and EC-measured carbon fluxes. In doing so, our study also paves a way to exemplar protocols to appropriately use EC time series and satellite-imagery VI time series together for investigating and monitoring crop carbon fluxes in future studies.

Concerning "suitability and developability": Following the scientific principle of occam's razor and in respect of the complexity of mechanistic crop models not outperforming clearly the simple linear regressions we argue that this study has its value by showing just that. The complex and heavy on input parameters and often assumptions models are not necessarily better than simpler models. By stating "suitability and developability" we acknowledge the unknown if different crop types/traits/environmental conditions require a higher resolution of linear models. But this was not the aim of this study. At the same time this means as well that

we acknowledge the research demand to address just these questions. We found one reference addressing the question of generalizability of the linear models (Juszczak et al. 2018) which argues that a generic single relationship between VI and C flux *can* be valid for a range of different crops.

Furthermore, we acknowledge the research demand for finding potential additional meteorological variables (in the conclusions) once this approach is applied at more sites and results indicate the necessity of including more explanatory variables. However, we do like to mention that additional variables also introduce additional sources of error/uncertainty at spatial scales because e.g. temperature at spatial scale will always be uncertain as well. However, this would have exceeded the scope of this manuscript and is topic of a follow-up study.

We will elaborate on these points in a revised manuscript.

The results/conclusions presented from this study did not seem to necessarily address the limitation of low data density of the flux observations, as the results were derived from one crop flux tower in Germany with seemingly low data density (compensating for better data quality).

The intention was not to address the low availability of flux observations but – amongst others – to leverage a cropland site with detailed flux measurements and meter-scale satellite images to achieve a detailed and rigorous analysis into the relationship between satellite-imagery VI and EC-measured carbon fluxes. In doing so, we also evaluate whether large-scale statistical approaches (as used for FLUXCOM) are actually justified when evaluated at plot scale.

Furthermore, this case study not only shines light on the great potential of satellite image-based VI for carbon flux estimation but also serves as a template how to appropriately use EC data stream and imagery time series jointly in later studies over more sites in different climate regions.

Moreover, regarding the high-resolution capability, are there any prior studies that utilized Vis from Landsat and Sentinel-2 for estimating carbon fluxes over croplands at meter scales?

Some prior studies use Landsat and/or Sentinel-2 images for estimating carbon fluxes over vegetated lands, mostly looking at GPP (e.g. Pabon-Moreno et al., 2022, Spinosa et al. 2023, Fu et al., 2014, Madugundu et al., 2017, Chen et al., 2010, Wolanin et al., 2019, Bazzi et al., 2024) but very few at NEE specifically over croplands. We will cite these prior studies in the discussion section. Our study uses meter-scale satellite images to also estimate NEE, which is of higher necessity and importance to calculate C budgets but is more challenging to estimate due to its indirect correlation with VIs.

In sum, the knowledge/technical gaps from prior studies and the novelty of this study can be better phrased. I would suggest conducting a more thorough literature review and rewording relevant text to highlight the advances of this study from prior studies (e.g., the examination of various VIs, the estimation of the carbon budget, the alignment of footprint between VIs and flux signals, and investigation of crop management signals from EC and VI data, from my understanding?).

Good point. We will elucidate and phrase better the novel aspects of this study in relation to more dedicated literature. To our knowledge there is only one study though dedicated at

assessing the impact of the alignment or not-alignment of EC with satellite footprints when estimating C fluxes (Kong et al. 2022). Furthermore, we will highlight better that our study fills the gap in exemplar protocols to appropriately use EC time series and satellite-imagery VI time series together for investigating and monitoring crop carbon fluxes in future studies.

Here are some minor comments:

Figures 2, 3: the time format on the x-axis is not super intuitive. Either change the date/time format or add a caption. What do the vertical lines in the dashed or dotted lines indicate in Figure 3? Add a few text labels to facilitate the comprehension of the discussion on page 18 (e.g., the timing for flowering on L273 and senescence on L276 - 278).

Yes, agreed. Same has been mentioned by RC1 and this formatting issue will be addressed as stated in the response to RC1: changing x-axis labels, add explanation of dashed and dotted lines to Fig. 3, adding horizonal segments indicating the timing of flowering and senescence.

L294-296: The poorer correlation between $R_{eco}$ and VIs seems as expected. Does that suggest additional meteorological variables describing the air and soil columns are needed?

This is probably true when focusing at Reco. We will add relevant words in the discussion.

There are a lot of numbers going into the result sections and tables. There could be better ways to present the data more intuitively, e.g., replacing Table 6 with a bar plot.

Yes, we will provide the information also as a figure. We would still like to add the table in the appendix as it can be frustrating when results are only shown as plots when actual numbers are needed for comparison or reference.

**References**

Bazzi, H., et al. (2024). "Assimilating Sentinel-2 data in a modified vegetation photosynthesis and respiration model (VPRM) to improve the simulation of croplands CO2 fluxes in Europe." International Journal of Applied Earth Observation and Geoinformation 127: 103666.

Chen, B., et al. (2010). "A data-model fusion approach for upscaling gross ecosystem productivity to the landscape scale based on remote sensing and flux footprint modelling." Biogeosciences 7(9): 2943-2958.

Fu, D., et al. (2014). "Estimating landscape net ecosystem exchange at high spatial–temporal resolution based on Landsat data, an improved upscaling model framework, and eddy covariance flux measurements." Remote Sensing of Environment 141: 90-104.

Huang, X., et al. (2019). "Evaluating the Performance of Satellite-Derived Vegetation Indices for Estimating Gross Primary Productivity Using FLUXNET Observations across the Globe." Remote Sensing 11(15): 1823.

Juszczak, R., Uździcka, B., Stróżecki, M., and Sakowska, K.: Improving remote estimation of winter crops gross ecosystem production by inclusion of leaf area index in a spectral model, PeerJ, 6, e5613, 10.7717/peerj.5613, 2018.

Kong, J., Ryu, Y., Liu, J., Dechant, B., Rey-Sanchez, C., Shortt, R., Szutu, D., Verfaillie, J., Houborg, R., and Baldocchi, D. D.: Matching high resolution satellite data and flux tower footprints improves their agreement in photosynthesis estimates, Agricultural and Forest Meteorology, 316, 108878, https://doi.org/10.1016/j.agrformet.2022.108878, 2022

Madugundu, R., et al. (2017). "Estimation of gross primary production of irrigated maize using Landsat-8 imagery and Eddy Covariance data." Saudi Journal of Biological Sciences 24(2): 410-420.

Pabon-Moreno, D. E., et al. (2022). "On the Potential of Sentinel-2 for Estimating Gross Primary Production." IEEE Transactions on Geoscience and Remote Sensing 60: 1-12.

Sims, D. A., et al. (2006). "On the use of MODIS EVI to assess gross primary productivity of North American ecosystems." Journal of Geophysical Research: Biogeosciences 111(G4).

Spinosa, A., et al. (2023). "Assessing the Use of Sentinel-2 Data for Spatio-Temporal Upscaling of Flux Tower Gross Primary Productivity Measurements." Remote Sensing 15(3): 562.

Tramontana, G., Ichii, K., Camps-Valls, G., Tomelleri, E., and Papale, D.: Uncertainty analysis of gross primary production upscaling using Random Forests, remote sensing and eddy covariance data, Remote Sensing of Environment, 168, 360-373, https://doi.org/10.1016/j.rse.2015.07.015, 2015.

Tramontana, G., Jung, M., Schwalm, C. R., Ichii, K., Camps-Valls, G., Ráduly, B., Reichstein, M., Arain, M. A., Cescatti, A., Kiely, G., Merbold, L., Serrano-Ortiz, P., Sickert, S., Wolf, S., and Papale, D.: Predicting carbon dioxide and energy fluxes across global FLUXNET sites with regression algorithms, Biogeosciences, 13, 4291-4313, 10.5194/bg-13-4291-2016, 2016.

Wolanin, A., et al. (2019). "Estimating crop primary productivity with Sentinel-2 and Landsat 8 using machine learning methods trained with radiative transfer simulations." Remote Sensing of Environment 225: 441-457.

---

## Author Response (AR1)

Dear Andrew Feldman,

thank you very much for handling our manuscript and giving us the opportunity to revise the manuscript - and for granting an extension for the revision process! Below you will find a point-by-point response to your comments and to the comments of the two reviewers. Responses are in blue.

First of all, we adjusted the copyright statement for Figure 1 and we did adjust the figure in Appendix B to "Figure B1" as for the remarks on the file validation. We had already submitted these changes for the first submission.

Relevant changes:

- Rephrasing of the introduction to better work out the novelty of the study
- Including relevant literature as pointed out by the second reviewer (previous studies on high-resolution satellite imagery used for NEE estimates) in the Introduction and "Results and Discussion" section.
- Figure adjustments
- Correcting grammar and orthography as pointed out by the first reviewer and where found by ourselves.
- All minor revisions requested by each reviewer except changing tables to figures.

Thank you for your patience. We've received two reviews. While both are positive, reviewer 2 questioned the novelty and the representativeness of the results beyond the study region. I agree with these comments and think they should be addressed before publication.

I myself appreciate the approach because with more remote sensing instruments moving to higher resolution, applications of co-use of satellite data and field data need to be explored and extended more widely. A carbon flux application like this is certainly needed.

As in the authors' reply, it should be highlighted that the novelty lies in the very high resolution of the data and that such applications are becoming more possible with high resolution instruments (like OCO-3 and others upcoming (SBG), as least on the NASA side. Others from other satellite agencies can be mentioned).

We substantially rephrased the introduction to work out the highlights of the study according to the comments by reviewer 2, the high spatial resolution is only one. We also do mention current and upcoming satellite missions which provide high ground-resolution products.

Adding to the dialogue between the reviewer and authors, I am wondering if the authors should test whether a more complex regression model with more regression variables is necessary here. I understand the desire to see how much VI's alone explain the in-situ carbon fluxes. At least, the authors should motivate why only simple relationships with regressing VI only is appropriate.

The revisions of the introduction do now include the motivation why we are testing this simple relationship. As mentioned, it has indeed been our intention to evaluate how much VIs alone explain C flux dynamics. We appreciate the request to extend the approach to integrate more explanatory variables which is certainly the next step. However, the integration of more explanatory variables, also in conjunction with more complex methods, such as ecosystem models or data-model fusion approaches do not show decisively better results as presented in the discussion. We thus think that elucidating the explanatory gap would be subject of an extra study which can build nicely on the current one.

Reviewer 1:

Overview

This manuscript combines field scale EC data and satellite derived VIs to estimate the carbon flux over a local cropland in Europe. Several kinds of VIs are used to estimate NEE and GPP based on linear regression models. Their method is reasonably straightforward, and assumption is clearly stated. The results are well investigated from multiple aspects, which makes their conclusions more concrete. Their estimated values including uncertainties are fairly compared to previous works. The authors explain the strengths and weaknesses of their approach very well. Some figures and tables may be a little hard to read/follow to readers, though.

Thank you for the evaluation of our manuscript and for the constructive comments and suggestions! In the following we provide a point-by-point response to all comments and suggestions.

Followings are my minor comments and suggestions.

In the abstract, please clarify the size/location of the study field.

We added the information of location and size in the abstract.

L20: Please define "RMSE"

We spelled out RMSE = "root mean square error".

L79: "limits"

Corrected.

L86: Please define "NIR"

Defined.

L114: Please define "a.s.l."

Spelled out as "above sea level".

L134: Data "were" measured…

Corrected.

L141: For date expression, please be consistent ("5th March" or "23 August").

We revised the whole manuscript accordingly.

L164: Please define "SCL".

Defined.

L198: There is no "section 0" in this manuscript.

Deleted.

L222: It will help to put the total NEE, GPP and Reco for each growing season in Fig.2 as you did for the climate parameters.

Done.

L254: "apart from" Do you mean "except for"?

Revised.

L277-278: The VI observations are sparse compared to NEE/GPP, so it might be hard to tell "these VIs lag about 18 days behind the NEE signal for the senescence period of WR" from the Fig. 3. Please specify how you verify it.

We added an elaboration: 18 days is the average across VIs based on the observation that VI values had reached a plateau at the same time as of max. C uptake (= min NEE) and did not drop at the same time as NEE decreases. Another similarly high VI value is observed about 18 days later before the next VI value is much lower.

L308: Please define "lowess (should be LOWESS)".

Changed to "fitted line" as explained in the manual of the R-function used (chart.correlation of the R-package "PerformanceAnalytics").

L287-288: Why the signs of correlations are opposite of yours?

We added an explanatory note for the reader: "Please note, that Baldocchi (2008) reports GPP ('$F_A$') with a positive sign, not with a negative sign as we do and as in Baldocchi et al. (2015). Thus, his correlation value between GPP and Reco is positive as opposed to ours and in Baldocchi et al. (2015). The correlations can still be compared based on the absolute values."

L289: This sentence is hard to understand and seems not quite matching the results shown in Fig. 4.

Agreed. The sentence was split into two: "NEE correlated highly significant ($p<0.001$) with all VIs except MNDWI which showed a lower level of statistical significance ($p<0.01$). GPP correlated highly significant ($p<0.001$) with all VIs except MNDWI ($p<0.01$) and NDSVI ($p<0.05$) which both showed a lower level of statistical significance."

L321 and L324: "nir"--> please define and change to "NIR".

We changed all occurrences of "nir" to "NIR". The definition was given according to the previous comment by the reviewer (L86).

L322: "swir" --> please define and change to "SWIR".

We changed "swir" to "SWIR" and added the respective definition of "short-wave infrared".

L323: "amongst the highest" is not true. Do you mean "one of the highest"?

The sentence was changed to "This did not hold true for correlations between VIs and GPP, where NDWI showed the second highest correlation with GPP."

L383: Please define "E". "RMSE" does NOT "increased" (but improved).

Yes, "increased" was wrong. It was changed to "improved". "E" is defined in L186-189.

L405: It is not very clear that $R^2$-values are calculated between what parameters.

Agreed. Was changed to "Jaszczak et al. (2018) found a $R^2$-value for the linear regression between NDVI and SAVI, respectively, and GPP of WW of 0.56 and 0.59 ($p<0.0001$) respectively.

L422: "$\tau$" --> Do you mean "$\rho$"?

It is "τ" (Kendall tau rank correlation coefficient) as this is the correlation coefficient used in Wattenbach et al. (2010). We added a comment stressing a little more the different coefficients used.

L434-439: These sentences are hard to understand. "overestimation" and "underestimation" with respect to what? Your estimated results?

Over- and underestimations relative to the measured values at the simulated sites used in the cited publication. We modified the sentence to make that clear: "The model ensemble of Wattenbach et al. (2010) exhibited absolute simulation errors *of total annual C sums* ranging from an overestimation of *measured* seasonal C uptake of 204 g C m$^{-2}$ to an underestimation of *measured* C uptake of 217 g C m$^{-2}$ across *the* 5 WW site-years and 3 crop models *used in their study*."

L435: "204" and "217" are for cumulative C flux, right? Please mention that and specify the time period.

Correct. These values are annual sums. We added this clarification (see response to previous comment).

L440: "85%" --> "85.66%".

Agreed. Was changed.

L442: Please add ", respectively" after "(17.93%)".

Was added.

L443: "1.35 and 1.36" --> "1.36 and 1.35". "(Table 6)" --> "(Table 5)".

Yes, thanks for spotting this! Was changed.

L451: Please add ", respectively" after "2021/2022,".

Done.

L452-453: This sentence is a little hard to understand. I do not see how it is related to Table 3. Where the number "311 g C m$^{-2}$" come from?

"311 g C m$^{-2}$" is the difference of total growing season NEE values of 2020/2021 and 2021/2022 respectively, after converting from t C ha$^{-1}$ to g C m$^{-2}$. The sentence was changed to "Further, estimation-uncertainty is smaller than the difference of total growing season NEE between the two WW growing periods 2020/2021 and 2021/2022, respectively, which was 311 g C m$^{-2}$ (i.e. 3.11 t C ha$^{-1}$, Table 3)."

L467: Please add "," after "harvest".

Thanks, done.

In sections 3.5.2 and 3.5.3, more detailed explanations of the models are needed such as the model simulation years (is there any inter-annual variability)? Readers might understand better if section 3.5.3 starts with explanation of Table 6, and then give comparisons with references.

We first added an additional sentence to section 2.7 introducing the reader to the comparison of our results to results of simulation studies by mechanistic crop models. We started section 3.5.3 with the introduction to Table 6. We added the information of model simulation years and inter-annual variability from the simulation studies in section 3.5.2.

Figure 1: It is a little too hard to understand what the percentage of "cumulative FP area" means. How was it calculated? It is not clear the definitions of "main field" and "area of interest".

We changed "FP area" to "source area" in the caption of Figure 1 and guide the reader to Appendix B for more detailed information. We will there add a sentence in which the cumulative foot print area is explained as the source are of the fluxes, and the isolines denote the cumulative contribution of that area to the flux signal over the measurement period. We removed "area of interest" as it is synonymously used for main field and occurred only once (Appendix B, L551).

Figure 2 (The top panel): Please specify the duration of "flowering of WR". (the first 2 top panels): please explain the gray dots and black solid lines in the figure caption. Also, please differentiate the GPP and Reco by using different color or line type. I do not see the "urea spreading" in 2020/2021 listed in Table 1.

We added a horizontal segment to show the length of WR flowering and added the duration dates in the caption. We added an explanation of the gray dots and lines also in the caption. We used two different line and point types for GPP and Reco. We verified the "urea spreading" event with the farmer. First, it was labeled wrong and it actually was an organic fertilizer application. That was corrected in the figure. Second, we added in the management table the information, that urea applications are sometimes replaced by organic fertilizers - as explained by the farmer (when organic fertilizer applications become temporarily cheaper than mineral fertilizer).

Please give x-axis title as "Time" and do not use the numbers such as 5.3, 1.5, 1.7… which are hard to understand. You could show the dates (i.e. 05 Mar 2020) only for the start and end of the growing season or so. Also, it may be helpful for readers to put the x-tick values at the top of the figure, too.

X-axis title has been changed to "Time" and the date format has been changed to 03 Mar 2020 etc. giving only Jan und July in the following. X-ticks are added to the top of the figure.

Figure 3: Y-title for VIs should be "(-1)*VIs"? There is no explanation for blue dots. Again, the x-axis' thick name of "5.3, 1.6…" are hard to follow. Please give monthly tick marks and label them properly. Also, it would be easier for readers to compare the time series of NEE/GPP and VIs if they have a common zero (horizontal) line. Please add legends for the vertical broken- and dotted- lines.

Good point to label the secondary y-axis with (-1)*VI. Done. We added the explanation for the blue dots and changed the labeling of the x-axis. We added a legend for the vertical broken- and dotted- lines. We decide against a common 0 line because the range of the VIs differ quite a lot and readability would not improve when fixing a common 0 line.

Figure 5: Please put an x-title and change the x-tick name format.

We added the x-title and changed the format of the x-tick names to align with the other figures.

Table 1: What is "ca."?

Ca. stands for "circa". We revised to "approx."

Table 3: Please state the unit clearly.

All values are given in t C ha$^{-1}$ season$^{-1}$. It is clearly mentioned in the caption.

Table 4: Please explain what the bold and italic numbers indicate either in the caption or main text. Why are $\rho$-values not shown here (just wonder) as in Table 5? It could be better to make the line thicker between different growing periods (in Table 5, also).

We added the explanation of numbers in bold and italic in the caption. Bold indicate best performing VIs within a group and italic indicate worst performing VIs within a group. We explain in the caption that all values are significant to the level of 0.001 except where indicated with '**'. Thus, we avoid printing '***' in each cell. That holds true for tables 4 and 5. We improved readability by thickening the lines between the groups.

Table 5: Please explain what the bold numbers indicate.

We added the explanation of numbers in bold and italic in the caption as for Table 4.

Table 6: the top of the 3rd column should be "NEE and GPP estimated [g C m-2]". You could add "*mean*" to Table 4 and 5 as well because it was mentioned in the main text. What does the "**" indicate? Again, please explain about the bolded numbers (it seems they are mentioned in the next, but not all of them are bolded).

Thanks for spotting the wrong labeling of column 3. We revised to "Estimates". We added rows for mean values in table 4, 5 and 6. '**' actually has no meaning and was removed.

S1 The third paragraph: "Reco … but showed no response the grubbing events". Is this true? It seems the Reco increased between the first and second grubbing events. The last (fifth) paragraph, "GPP and Reco were both lower …": because the signs of GPP and Reco are opposite, "lower" is a little confusing. Could be better to say "smaller".

We checked in more detail on the response of Reco to soil cultivation events. There is no response detectible. The increase between the first and second soil cultivation event in 2020 is complementary to the increase in GPP due to regrowth and thus more root exudates leading to faster turnover and higher heterotrophic respiration, and higher autotrophic respiration. We changed "lower" to "smaller". Thanks.

Gottschalk et al. present a high-resolution investigation of the vegetation indices against the carbon fluxes from a tower in the cropland. They processed the flux observations thoroughly to ensure the data quality and estimated the fluxes based on linear models using VIs, which were then compared with crop models. They showed that observed fluxes correlate well with the field management actions, phenological development of crops, and some Sentinel-2-derived VIs (especially for GPP). They concluded with discussions of the limitations of their analyses (e.g., adopting a simple linear model).

Thank you for evaluating our manuscript and for providing constructive and helpful comments and suggestions. In the following we address all points raised by a point-by-point response.

Major comment:

In general, the data processing is thorough, and the results are well presented. Leveraging the high-resolution feature from Sentinel-2 data, their EC-based carbon fluxes and Vis are all matched to the plot scale, which addressed the common spatial mismatch issue in prior studies.

Despite the promising flux-VI relationship presented, my major concern lies in the novelty of this study and the representativeness of such flux-VI relationships across crop traits and climates (despite that the authors acknowledged till the very end). The authors suggested that their analyses indicate "the suitability and developability of the proposed approach to monitor cropland C exchange with satellite derived Vis*"* (L25-26) and acknowledged the current challenges in flux estimates being the low number of EC towers in croplands (L79-81). When considering croplands over different climates/regions, how robust is the use of a linear model to predict carbon fluxes? Would involving additional meteorological variables be more helpful (which is relative to statements on L490-491)?

Sims et al., 2006 and Huang et al. 2019 have demonstrated successfully the usefulness of linear models to estimate GPP across ecosystems. We now take this a step further to estimate NEE with linear regression models for a range of high-resolution satellite-derive VIs which has not yet been done before for croplands and only for a limited number of grassland sites. The question of robustness can only be assessed with a greater number of sites for which this study presents a template, also addressing the issue of uncertainty coming from spatially mismatched signals. Further, studies for flux estimation using more complex approaches in conjunction with satellite data do not as such achieve better results and the additional value of meteorological parameters at the level or accuracy of our results has not been shown. We thus argue that this study is worthwhile showing the high potential of employing high-resolution satellite imagery and delivering a template for further investigation along this path. We do acknowledge of course the limitation of this study being confined to one site. But this is clearly addressed throughout the manuscript and provides the bases for follow-up studies.

The results/conclusions presented from this study did not seem to necessarily address the limitation of low data density of the flux observations, as the results were derived from one crop flux tower in Germany with seemingly low data density (compensating for better data quality).

The intention was not to address the low availability of flux observations but – amongst others – to leverage a cropland site with detailed flux measurements and meter-scale satellite images to achieve a detailed and rigorous analysis into the relationship between satellite-imagery VI and EC-measured carbon fluxes. In doing so, we also evaluate whether large-scale statistical approaches are actually justified when evaluated at plot scale.

Moreover, regarding the high-resolution capability, are there any prior studies that utilized Vis from Landsat and Sentinel-2 for estimating carbon fluxes over croplands at meter scales?

Some prior studies use Landsat and/or Sentinel-2 images for estimating carbon fluxes over vegetated lands, mostly looking at GPP (e.g. Pabon-Moreno et al., 2022, Spinosa et al. 2023, Fu et al., 2014, Madugundu et al., 2017, Chen et al., 2010, Wolanin et al., 2019, Bazzi et al., 2024) but very few are looking at NEE specifically over croplands. We cited these prior studies in the introduction and compared our results to these studies when appropriate in section 3.5.2.

In sum, the knowledge/technical gaps from prior studies and the novelty of this study can be better phrased. I would suggest conducting a more thorough literature review and rewording relevant text to highlight the

advances of this study from prior studies (e.g., the examination of various VIs, the estimation of the carbon budget, the alignment of footprint between VIs and flux signals, and investigation of crop management signals from EC and VI data, from my understanding?).

We rephrased the introduction substantially to better work out the novelty of this study and we compared our results to additional studies estimating C fluxes with satellite-data-model-fusion approaches. These comparisons clearly show that a linear approach is already capable of achieving high levels of accuracy. Thus, the main take-home message is, that a rigorous use of state-of-the-art data products can be leveraged to produce a robust first estimate of cropland C fluxes without the need of additional meteorological variables or plant physiological parameters. Improvements are always possible which is the motivation for a follow-up study!

Here are some minor comments:

Figures 2, 3: the time format on the x-axis is not super intuitive. Either change the date/time format or add a caption. What do the vertical lines in the dashed or dotted lines indicate in Figure 3? Add a few text labels to facilitate the comprehension of the discussion on page 18 (e.g., the timing for flowering on L273 and senescence on L276 - 278).

We amended figures 3-5 and changed the format of the x-axis and added legend items and further explanations in the captions.

L294-296: The poorer correlation between $R_{eco}$ and VIs seems as expected. Does that suggest additional meteorological variables describing the air and soil columns are needed?

We discussed the lower correlation and its potential causes in section 3.3 "Correlations between daily C fluxes and VIs".

There are a lot of numbers going into the result sections and tables. There could be better ways to present the data more intuitively, e.g., replacing Table 6 with a bar plot.

Yes, we have been considering changing tables to figures. However, we do prefer to stick with the tables as the information is concrete and readers can actually use numbers for comparison and referencing.